# Transfer Learning in $\ell_1$ Regularized Regression: Hyperparameter Selection Strategy based on Sharp Asymptotic Analysis

**Koki Okajima**                                                    *darjeeling@g.ecc.u-tokyo.ac.jp*
*Graduate School of Science, Department of Physics*
*The University of Tokyo*

**Tomoyuki Obuchi**                                                  *obuchi@i.kyoto-u.ac.jp*
*Graduate School of Informatics, Department of Informatics*
*Kyoto University*

**Reviewed on OpenReview:** *https://openreview.net/forum?id=ccuOM3nmlF*

## Abstract

Transfer learning techniques aim to leverage information from multiple related datasets to enhance prediction quality against a target dataset. Such methods have been adopted in the context of high-dimensional sparse regression, and some Lasso-based algorithms have been invented: Trans-Lasso and Pretraining Lasso are such examples. These algorithms require the statistician to select hyperparameters that control the extent and type of information transfer from related datasets. However, selection strategies for these hyperparameters, as well as the impact of these choices on the algorithm's performance, have been largely unexplored. To address this, we conduct a thorough, precise study of the algorithm in a high-dimensional setting via an asymptotic analysis using the replica method. Our approach reveals a surprisingly simple behavior of the algorithm: Ignoring one of the two types of information transferred to the fine-tuning stage has little effect on generalization performance, implying that efforts for hyperparameter selection can be significantly reduced. Our theoretical findings are also empirically supported by applications on real-world and semi-artificial datasets using the IMDb and MNIST datasets, respectively.

## 1 Introduction

The increasing availability of large-scale datasets has led to an expansion of methods such as pretraining and transfer learning (Torrey & Shavlik, 2010; Weiss et al., 2016; Zhuang et al., 2021), which exploit similarities between datasets obtained from different but related sources. By capturing general patterns and features, which are assumed to be shared across data collected from multiple sources, one can improve the generalization performance of models trained for a specific task, even when the data for the task is too limited to make accurate predictions. Transfer learning has demonstrated its efficacy across a wide range of applications, such as image recognition (Zhu et al., 2011), text categorization (Zhuang et al., 2011), bioinformatics (Petegrosso et al., 2016), and wireless communications (Pan et al., 2011; Wang et al., 2021). It is reported to be effective in high-dimensional statistics (Banerjee et al., 2020; Takada & Fujisawa, 2020; Bastani, 2021; Li et al., 2021), with clinical analysis (Turki et al., 2017; Hajiramezanali et al., 2018; McGough et al., 2023; Craig et al., 2024) being a major application area due to the scarcity of patient data and the need to comply with privacy regulations. In such applications, one must handle high-dimensional data, where the number of features greatly exceeds the number of samples. This has sparked interest in utilizing transfer learning to sparse high-dimensional regression, which is the standard method for analyzing such data.

A classical method for statistical learning under high-dimensional settings is Lasso (Tibshirani, 1996), which aims at simultaneously identifying the sparse set of features and their regression coefficients that are most

relevant for predicting the target variable. Its simplicity and interpretability of the results, as well as its convex property as an optimization problem, has made Lasso a popular choice for sparse regression tasks, leading to a flurry of research on its theoretical properties (Candes & Tao, 2006; Zhao & Yu, 2006; Meinshausen & Bühlmann, 2006; Wainwright, 2009; Bellec et al., 2018).

Modifications to Lasso have been proposed to incorporate and take advantage of auxiliary samples. The data-shared Lasso (Gross & Tibshirani, 2016) and stratified Lasso (Ollier & Viallon, 2017) are both methods based on multi-target learning, where one solves multiple regression problems with related data in a single fitting procedure. Two more recent approaches are Trans-Lasso (Bastani, 2021; Li et al., 2021) and Pretraining Lasso (Craig et al., 2024), both of which require two-stage regression procedures. Here, the first stage is focused on identifying common features shared across multiple related datasets, while the second stage is dedicated to fine-tuning the model on a specific target dataset. To simplify the explanation, we introduce a generalized algorithm that encompasses both methods, which is referred to as *Generalized Trans-Lasso* hereafter. Given a set of datasets of size $K$, $\mathcal{D} = \{\mathcal{D}^{(k)}\}_{k=1}^K$, each consisting of an observation vector of dimension $M_k$, $\boldsymbol{y}^{(k)} \in \mathbb{R}^{M_k}$, and its corresponding covariate matrix $\boldsymbol{A}^{(k)} \in \mathbb{R}^{M_k \times N}$, generalized Trans-Lasso performs the following two steps:

1. Pretraining procedure: In the first stage, all datasets are used to identify a common feature vector $\hat{\boldsymbol{x}}^{(\mathsf{1st})}$ which captures shared patterns and relevant features across the datasets:

$$\hat{\boldsymbol{x}}^{(\mathsf{1st})} = \underset{\boldsymbol{x} \in \mathbb{R}^N}{\arg\min} \, \mathcal{L}^{(\mathsf{1st})}(\boldsymbol{x}; \mathcal{D}), \qquad \mathcal{L}^{(\mathsf{1st})}(\boldsymbol{x}; \mathcal{D}) = \frac{1}{2} \sum_{k=1}^K \|\boldsymbol{y}^{(k)} - \boldsymbol{A}^{(k)} \boldsymbol{x}\|_2^2 + \lambda_1 \|\boldsymbol{x}\|_1. \tag{1}$$

2. Fine-tuning: Here, the interest is only on a specific target dataset, which is designated by $k = 1$ without loss of generality. The second stage incorporates the learned feature vector's configuration and its support set into a modified sparse regression model. This is done by first offsetting the observation vector $\boldsymbol{y}^{(1)}$ by a common prediction vector deduced from the first stage, i.e. $\boldsymbol{A}^{(1)} \hat{\boldsymbol{x}}^{(\mathsf{1st})}$. Further, the penalty factor is also modified such that variables not selected in the first stage is penalized more heavily than those selected. The above procedure is summarized by the following optimization problem:

$$\hat{\boldsymbol{x}}^{(\mathsf{2nd})} = \underset{\boldsymbol{x} \in \mathbb{R}^N}{\arg\min} \, \mathcal{L}^{(\mathsf{2nd})}(\boldsymbol{x}; \hat{\boldsymbol{x}}^{(\mathsf{1st})}, \mathcal{D}^{(1)}),$$

$$\mathcal{L}^{(\mathsf{2nd})}(\boldsymbol{x}; \hat{\boldsymbol{x}}^{(\mathsf{1st})}, \mathcal{D}^{(1)}) = \frac{1}{2} \|\boldsymbol{y}^{(1)} - \kappa \boldsymbol{A}^{(1)} \hat{\boldsymbol{x}}^{(\mathsf{1st})} - \boldsymbol{A}^{(1)} \boldsymbol{x}\|_2^2 + \sum_{i=1}^N \left( \lambda_2 + \Delta\lambda \mathbb{I}[\hat{x}_i^{(\mathsf{1st})} = 0] \right) |x_i|,$$

(2)

where $\mathbb{I}$ is the indicator function. Here, $\kappa, \Delta\lambda \geq 0$ are hyperparameters that determine the extent of information transfer from the first stage.

It is crucial to note that the performance of this algorithm is highly dependent on the choice of hyperparameters, which control the strength of knowledge transfer. Suboptimal hyperparameter selection can lead to insufficient knowledge transfer, or negative transfer (Pan et al., 2011), where the model's performance is degraded by the transfer of adversarial information. Thus, the choice of appropriate hyperparameters should be addressed with equal importance as the algorithm itself. In fact, a distinctive difference between Trans-Lasso and Pretraining Lasso is whether one should inherit support information from the first stage or not. While this adds an additional layer of complexity to hyperparameter selection, it also provides an opportunity to inherit significant information from the first stage, if the support of the feature vectors between the sources are similar. Empirical investigation of the qualitative effects of hyperparameters on the algorithm's performance can be computationally intensive, especially in high-dimensional settings.

To address the issue of hyperparameter selection without resorting to extensive empirical research, one can turn to the field of statistical physics, which provides a set of tools to analyze the performance of high-dimensional statistical models in a theoretical manner. While these theoretical studies are restricted to inevitably simplified assumptions on the statistical properties of the dataset, they often provide valuable

qualitative insights generalizable to more realistic settings. The replica method (Mezard et al., 1986; Charbonneau et al., 2023), in particular, has been widely used to precisely analyze high-dimensional statistical models including Lasso (Kabashima et al., 2009; Rangan et al., 2009; Obuchi & Kabashima, 2016; Takahashi & Kabashima, 2018; Obuchi & Kabashima, 2019; Okajima et al., 2023), and has been shown to provide accurate predictions in many cases.

In this work, we conduct a sharp asymptotic analysis of the generalized Trans–Lasso's performance under a high-dimensional setting using the replica method. Our study reveals a simple heuristic strategy to select hyperparameters. More specifically, it suffices to use either the support information or the actual value of the feature vector $\hat{\boldsymbol{x}}^{(\mathsf{1st})}$ obtained from the pretraining stage to achieve near-optimal performance after fine-tuning. Our theoretical findings are complemented through empirical experiments on the real-world IMDb dataset (Maas et al., 2011) and semi-artificial datasets derived from MNIST images (Deng, 2012).

## 1.1 Related Works

Trans-Lasso, first proposed by Bastani (2021) and further studied by Li et al. (2021), is a special case of the two-stage regression algorithm given by equations 1 and 2. In this variant, the second stage omits information transfer regarding the support of $\hat{\boldsymbol{x}}^{(\mathsf{1st})}$ (i.e., $\Delta\lambda = 0$) and sets $\kappa$ to unity. This method has demonstrated efficacy in enhancing the generalization performance of Lasso regression, and has since been extended to various regression models, such as generalized linear models (Tian & Feng, 2023; Li et al., 2024), Gaussian graphical models (Li et al., 2023), and robust regression (Sun & Zhang, 2023). Moreover, the method also embodies a framework to determine which datasets are adversarial in transfer learning, allowing one to avoid negative transfer. While the problem of dataset selection is a practically important issue in transfer learning, the present study will not consider this aspect, but rather focus on the algorithm's performance given a fixed set of datasets.

On the other hand, Pretraining Lasso controls both $\kappa$ and $\Delta\lambda$ using a tunable interpolation parameter $s \in [0, 1]$ as

$$\kappa = 1 - s, \qquad \Delta\lambda = \frac{1-s}{s}\lambda_2. \tag{3}$$

This formulation allows for a continuous transition between two extremes: Setting $s = 1$ reduces the problem to a standard single-dataset regression, while $s = 0$ reduces to regression constrained on the support of $\hat{\boldsymbol{x}}^{(\mathsf{1st})}$. Note that there is no choice of $s$ which reduces Pretraining Lasso to Trans-Lasso.

Previous research has not yet established a clear understanding of the optimal hyperparameter selection for these methods. Moreover, the introduction of additional hyperparameters, such as those proposed in this study, has been largely unexplored in existing literature, possibly due to the computational demands involved. A more thorough investigation into the performance of these methods is necessary to reevaluate current, potentially suboptimal hyperparameter choices and to provide a more comprehensive understanding of the algorithm's behavior.

The analysis of our work is based on the replica method, which is a non-rigorous mathematical tool used in a wide range of theoretical studies, such as those on spin-glass systems and high-dimensional statistical models. The results from such analyses have been shown to be consistent with empirical results, and in some cases later proved by mathematically rigorous methods, such as approximate message passing theory (Bayati & Montanari, 2011; Javanmard & Montanari, 2013), adaptive interpolation (Barbier et al., 2019; Barbier & Macris, 2019), Gordon comparison inequalities (Stojnic, 2013; Thrampoulidis et al., 2018; Miolane & Montanari, 2021), and second-order Stein formulae (Bellec & Zhang, 2021; Bellec & Shen, 2022). The sharpness of the formulae obtained in these approaches have also be exploited to gain insight into the effect of hyperparameters on the performance of learning methods, such as bagging estimators Takahashi (2023); Koriyama et al. (2024), stochastic algorithms (Takahashi, 2024; Lou et al., 2024), and robust regression (Vilucchio et al., 2024). The application of the replica method to multi-stage procedures in machine learning has also been done in the context of knowledge distillation (Saglietti & Zdeborová, 2022), self-training (Takahashi, 2024), and iterative algorithms (Okajima & Takahashi, 2024). Our analysis can be seen as an extension of these works to this generalized Trans-Lasso algorithm.

## 2 Problem Setup

As given in the Introduction, we consider a set of $K$ datasets, each consisting of an observation vector $\boldsymbol{y}^{(k)} \in \mathbb{R}^{M_k}$ and covariate matrices $\boldsymbol{A}^{(k)} \in \mathbb{R}^{M_k \times N}$. The observation vector $\boldsymbol{y}^{(k)}$ is assumed to be generated from a linear model with Gaussian noise, i.e.

$$\boldsymbol{y}^{(k)} = \boldsymbol{A}^{(k)} \boldsymbol{r}^{(k)} + \boldsymbol{e}^{(k)}, \qquad \boldsymbol{e}^{(k)} \sim \mathcal{N}(0, (\sigma^{(k)})^2 \boldsymbol{I}_{M_k}). \tag{4}$$

Here, $\boldsymbol{e}^{(k)} \in \mathbb{R}^{M_k}$ is a Gaussian-distributed noise term, while $\boldsymbol{r}^{(k)} \in \mathbb{R}^N$ is the true feature vector of the underlying linear model. We consider the common and individual support model (Craig et al., 2024), where $\{\boldsymbol{r}^{(k)}\}_{k=1}^K$ is assumed to have overlapping support. More concretely, let $\{\mathcal{I}(k)\}_{k=0}^K$ be a set of disjoint indices, where $\mathcal{I}(k) \subset \{1, 2, \ldots, N\}$, $|\mathcal{I}(k)| = N_k$, for all $k = 0, \cdots, K$, and $\sum_{k=0}^K N_k \leq N$. Then, $\boldsymbol{r}^{(k)}$ is given by

$$\boldsymbol{r}_{\mathcal{I}(0)}^{(k)} = \boldsymbol{x}_\star^{(0)} \in \mathbb{R}^{N_0}, \qquad \boldsymbol{r}_{\mathcal{I}(k)}^{(k)} = \boldsymbol{x}_\star^{(k)} \in \mathbb{R}^{N_k}, \qquad \boldsymbol{r}_{\backslash \{\mathcal{I}(k) \cup \mathcal{I}(0)\}}^{(k)} = \boldsymbol{0} \in \mathbb{R}^{N - N_k - N_0}, \tag{5}$$

and

$$\boldsymbol{y}^{(k)} = \boldsymbol{A}_{\mathcal{I}(0)}^{(k)} \boldsymbol{x}_\star^{(0)} + \boldsymbol{A}_{\mathcal{I}(k)}^{(k)} \boldsymbol{x}_\star^{(k)} + \boldsymbol{e}^{(k)}. \tag{6}$$

Here, $\boldsymbol{A}_{\mathcal{I}}^{(k)}$ denotes the submatrix of $\boldsymbol{A}^{(k)}$ with columns indexed by $\mathcal{I}$, $\boldsymbol{r}_{\mathcal{I}}^{(k)}$ denotes the subvector of $\boldsymbol{r}^{(k)}$ with indices in $\mathcal{I}$, and $\backslash \mathcal{I}$ is the shorthand of $\{1, 2, \ldots, N\} \backslash \mathcal{I}$. Therefore, each linear model is always affected by the variables in the common support set $\mathcal{I}(0)$, with additional variables in the unique support set $\mathcal{I}(k)$ for each dataset $k = 1, \cdots, K$. We refer to $\boldsymbol{x}_\star^{(0)}$ as the common feature vector, while $\boldsymbol{x}_\star^{(k)}$ as the unique feature vector for class $k$. The implicit aim of Trans-Lasso can be seen as estimating the common feature vector $\boldsymbol{x}^{(0)}$ in the first stage, treating the unique feature vectors in each class as noise. On the other hand, the second stage is dedicated to estimating the unique feature vector $\boldsymbol{x}_\star^{(1)}$ given a noisy, biased estimator of the common feature vector $\boldsymbol{x}_\star^{(0)}$, i.e. $\hat{\boldsymbol{x}}^{(\mathsf{1st})}$.

We evaluate the performance of the generalized Trans-Lasso using the generalization error in each stage, which is defined as the expected mean square error of the model prediction made against a set of new observations $\tilde{\mathcal{D}} = \{(\tilde{\boldsymbol{y}}^{(k)}, \tilde{\boldsymbol{A}}^{(k)})\}_{k=1}^K$ where $\tilde{\boldsymbol{y}}^{(k)} \in \mathbb{R}^{M_k}$ and $\tilde{\boldsymbol{A}}^{(k)} \in \mathbb{R}^{M_K \times N}$:

$$\epsilon^{(\mathsf{1st})} = \frac{1}{N} \sum_{k=1}^K \mathbb{E}_{\mathcal{D}, \tilde{\mathcal{D}}} \left[ \| \tilde{\boldsymbol{y}}^{(k)} - \tilde{\boldsymbol{A}}^{(k)} \hat{\boldsymbol{x}}^{(\mathsf{1st})} \|_2^2 \right], \tag{7}$$

for the first stage, while the generalization error for the second stage, with specific interest on class $k = 1$ without loss of generality, is given by

$$\epsilon^{(\mathsf{2nd})} = \frac{1}{N} \mathbb{E}_{\mathcal{D}, \tilde{\mathcal{D}}} \left[ \| \tilde{\boldsymbol{y}}^{(1)} - \tilde{\boldsymbol{A}}^{(1)} (\kappa \hat{\boldsymbol{x}}^{(\mathsf{1st})} + \hat{\boldsymbol{x}}^{(\mathsf{2nd})}) \|_2^2 \right]. \tag{8}$$

The objective of our analysis is to determine the behavior of the generalization error given the choice of hyperparameters $(\lambda_1, \lambda_2, \kappa, \Delta\lambda)$.

Note that the definition of $\epsilon^{(\mathsf{1st})}$ in equation 7 is appropriate for the case where the new data is redrawn, including the covariate matrix, from the common and independent support model with the same hyperparameters as the training data. In other scenarios, different definitions might be more suitable. For instance, if we consider the case where only one datapoint is newly observed equally for each class $k \in \{1, 2, \ldots, K\}$, the generalization error should be defined to have equal weight over all the classes, which is different from ours. In the analytical framework discussed below, that case can also be treated: The resultant formula of the generalization error would become equation 21 with all $\alpha^{(k)}$ fixed equal. This would give quantitatively different results from those based on equation 7, yielding another interesting situation. However, the investigation of that situation requires another detailed computation which we leave as future work.

**High-dimensional setup.** We focus on the asymptotic behavior in a high-dimensional limit, where the length of the common and unique feature vectors, as well as the size of the datasets for each class, tend to infinity at the same rate, i.e.

$$\frac{N_0}{N} \to \pi^{(0)} = O(1), \qquad \frac{N_k}{N} \to \pi^{(k)} = O(1), \qquad \frac{M_k}{N} \to \alpha^{(k)} = O(1), \quad k = 1, \cdots, K, \qquad \text{as } N \to \infty. \tag{9}$$

Also, define the proportion of the number of variables not included in any of the support sets as $\pi^{\text{(neg.)}} = 1 - \sum_{k=0}^{K} \pi^{(k)}$. The covariate matrices are all assumed to have i.i.d. entries with zero mean and variance $1/N$, while the true feature vectors are also assumed to have i.i.d. standard Gaussian entries. While such a setup is a drastic simplification of real-world scenarios, it should be noted that these conditions can be further generalized to more complex settings. For instance, the analysis can be further extended to the case where the random matrix $\boldsymbol{A}^{(k)}$ inherits rotationally invariant structure (Vehkaperä et al., 2016; Takahashi & Kabashima, 2018), or to the case where the observations are corrupted by heavy-tailed noise (Adomaityte et al., 2023).

The notations used in this paper are summarized in table 1.

| Notation | Description |
|---|---|
| $\mathcal{I}(0) \subset \{1, 2, \ldots, N\}$ | Support of features common across all classes |
| $\mathcal{I}(k) \subset \{1, 2, \ldots, N\}$ | Support of features unique to class $k = 1, \ldots, K$ |
| $N$ | Total number of features |
| $N_0$ | Size of common support, $|\mathcal{I}(0)|$ |
| $N_k$ | Size of independent support of class $k = 1, \ldots, K$, $|\mathcal{I}(k)|$ |
| $\boldsymbol{x}_\star^{(0)} \in \mathbb{R}^{N_0}$ | Common feature vector shared among all classes |
| $\boldsymbol{x}_\star^{(k)} \in \mathbb{R}^{N_k}$ | Unique feature vector exclusive to class $k = 1, \ldots, K$ |
| $\boldsymbol{y}^{(k)} \in \mathbb{R}^{M_k}$ | Observation vector for dataset $k = 1, \ldots, K$ |
| $\boldsymbol{e}^{(k)} \in \mathbb{R}^{M_k}$ | Gaussian noise term with i.i.d. elements of variance $(\sigma^{(k)})^2$ |
| $\boldsymbol{A}^{(k)} \in \mathbb{R}^{M_k \times N}$ | Covariate matrix for dataset $k = 1, \ldots, K$ |
| $\mathcal{D} = \{\mathcal{D}^{(k)}\}_{k=1}^{K}$ | Set of $K$ datasets. $\mathcal{D}^{(k)} = (\boldsymbol{y}^{(k)}, \boldsymbol{A}^{(k)})$ |
| $\tilde{\mathcal{D}} = \{\tilde{\mathcal{D}}^{(k)}\}_{k=1}^{K}$ | Set of $K$ test datasets with same distribution and size as $\mathcal{D}$ |
| $\hat{\boldsymbol{x}}^{\text{(1st)}}$ | First stage feature vector (pretraining) |
| $\hat{\boldsymbol{x}}^{\text{(2nd)}}$ | Second stage feature vector (fine-tuning) |
| $\lambda_1$ | First stage regularization parameter |
| $\lambda_2$ | Second stage base regularization parameter |
| $\kappa$ | Transfer coefficient to second stage |
| $\Delta\lambda$ | Additional regularization for features non-selected during pretraining |
| $\pi^{(0)}$ | Proportion of common features $N_0/N$ |
| $\pi^{(k)}$ | Proportion of unique features $N_k/N$ |
| $\pi^{\text{(neg.)}}$ | Proportion of features absent from all linear models, $1 - \sum_{k=0}^{K} \pi^{(k)}$ |
| $\alpha^{(k)}$ | Sample size ratio $M_k/N$ |

Table 1: Notations used in this paper.

## 3  Results of Sharp Asymptotic Analysis via the replica method

Here, we state the result of our replica analysis. The series of calculations to obtain our results is similar to the well-established procedure used in the analyses for multi-staged procedures (Saglietti & Zdeborová, 2022; Takahashi, 2024; Okajima & Takahashi, 2024). While a brief overview of the derivation is given in the next subsection, we refer the reader to Appendix A for a detailed calculation. Although our theoretical analysis lacks a standing proof from the non-rigorousness of the replica method, we conjecture that the results are exact in the limit of large $N$. This is supported numerically via experiments on finite-size systems, which will be shown to be consistent with the theoretical predictions.

In fact, since the first stage of the algorithm is a standard Lasso optimization problem under Gaussian design, it is already known that at least up to the first stage of the algorithm, the replica method yields the same formula derivable from Approximate Message Passing (Bayati & Montanari, 2011) or Convex Gaussian Minmax Theorem (CGMT) (Thrampoulidis et al., 2018; Miolane & Montanari, 2021). Moreover, CGMT can be further applied to multi-stage optimization problems with independent data in each stage (Chandrasekher et al., 2023). The extension of this proof to the second stage of our algorithm, which contains data dependence among the two stages, is an unsolved problem that we leave for future work.

### 3.1 Overview of the calculation

Here, we briefly outline the derivation of the generalization errors $\epsilon^{(\mathsf{1st})}$ and $\epsilon^{(\mathsf{2nd})}$ using the replica method. Readers not interested in the derivation may skip this subsection and proceed to the next subsection for the main results.

Given the training dataset $\mathcal{D} = \{\mathcal{D}^{(k)}\}_{k=1}^{K}$, with $\mathcal{D}^{(k)} = (\boldsymbol{A}^{(k)}, \boldsymbol{y}^{(k)})$, $k = 1, \cdots, K$, define the following joint probability measure for $\boldsymbol{x}_1 \in \mathbb{R}^N$ and $\boldsymbol{x}_2 \in \mathbb{R}^N$ as

$$P_{\beta_1, \beta_2}(\boldsymbol{x}_1, \boldsymbol{x}_2; \mathcal{D}) = \mathcal{Z}^{-1} w_{\beta_1, \beta_2}(\boldsymbol{x}_1, \boldsymbol{x}_2; \mathcal{D}),$$

$$w_{\beta_1, \beta_2}(\boldsymbol{x}_1, \boldsymbol{x}_2; \mathcal{D}) = \exp\left[ -\beta_1 \mathcal{L}^{(\mathsf{1st})}(\boldsymbol{x}_1; \mathcal{D}) - \beta_2 \mathcal{L}^{(\mathsf{2nd})}(\boldsymbol{x}_2; \boldsymbol{x}_1, \mathcal{D}^{(1)}) \right],$$

(10)

where $\mathcal{Z} = \int d\boldsymbol{x}_1 d\boldsymbol{x}_2 w_{\beta_1, \beta_2}(\boldsymbol{x}_1, \boldsymbol{x}_2; \mathcal{D})$ is the normalization constant. Under this measure, the generalization error $\epsilon^{(\mathsf{1st})}$ and $\epsilon^{(\mathsf{2nd})}$ can be expressed as

$$\epsilon^{(\mathsf{1st})} = \mathbb{E}_{\mathcal{D}, \tilde{\mathcal{D}}}\left[ \mathcal{Z}^{-1} \int d\boldsymbol{x}_1 d\boldsymbol{x}_2 w_{\beta_1, \beta_2}(\boldsymbol{x}_1, \boldsymbol{x}_2; \mathcal{D}) \sum_{k=1}^{K} \|\tilde{\boldsymbol{y}}^{(k)} - \tilde{\boldsymbol{A}}^{(k)} \boldsymbol{x}_1\|_2^2 \right]$$

$$= \lim_{\beta_2 \to \infty} \lim_{\beta_1 \to \infty} \lim_{t_1 \to 0} \frac{1}{N} \mathbb{E}_{\mathcal{D}, \tilde{\mathcal{D}}}\left[ \frac{\partial}{\partial t_1} \log \int d\boldsymbol{x}_1 d\boldsymbol{x}_2 w_{\beta_1, \beta_2}(\boldsymbol{x}_1, \boldsymbol{x}_2; \mathcal{D})\, e^{t_1 \sum_{k=1}^{K} \|\tilde{\boldsymbol{y}}^{(k)} - \tilde{\boldsymbol{A}}^{(k)} \boldsymbol{x}_1\|_2^2} \right], \quad (11)$$

$$\epsilon^{(\mathsf{2nd})} = \mathbb{E}_{\mathcal{D}, \tilde{\mathcal{D}}}\left[ \mathcal{Z}^{-1} \int d\boldsymbol{x}_1 d\boldsymbol{x}_2 w_{\beta_1, \beta_2}(\boldsymbol{x}_1, \boldsymbol{x}_2; \mathcal{D}) \|\tilde{\boldsymbol{y}}^{(1)} - \tilde{\boldsymbol{A}}^{(1)}(\kappa \boldsymbol{x}_1 + \boldsymbol{x}_2)\|_2^2 \right]$$

$$= \lim_{\beta_2 \to \infty} \lim_{\beta_1 \to \infty} \lim_{t_2 \to 0} \frac{1}{N} \mathbb{E}_{\mathcal{D}, \tilde{\mathcal{D}}}\left[ \frac{\partial}{\partial t_2} \log \int d\boldsymbol{x}_1 d\boldsymbol{x}_2 w_{\beta_1, \beta_2}(\boldsymbol{x}_1, \boldsymbol{x}_2; \mathcal{D})\, e^{t_2 \|\tilde{\boldsymbol{y}}^{(1)} - \tilde{\boldsymbol{A}}^{(1)}(\kappa \boldsymbol{x}_1 + \boldsymbol{x}_2)\|_2^2} \right]. \quad (12)$$

The first lines of equations 11 and 12 stem from the fact that, in the successive limit of $\beta_1 \to \infty$ followed by $\beta_2 \to \infty$, the joint probability measure $P_{\beta_1, \beta_2}(\boldsymbol{x}_1, \boldsymbol{x}_2; \mathcal{D})$ concentrates around $(\hat{\boldsymbol{x}}^{(\mathsf{1st})}, \hat{\boldsymbol{x}}^{(\mathsf{2nd})})$, the solution to the two-stage Trans-Lasso procedure. The second lines of equations 11 and 12 can be derived by simply differentiating the logarithm, and exchanging the data average and the limit of $t_1 \to 0$ or $t_2 \to 0$. These expressions do not require the computation of the inverse normalization constant $\mathcal{Z}^{-1}$, which is difficult to perform in general. Still, it requires one to evaluate the average of the logarithm of the integral of $w_{\beta_1, \beta_2}(\boldsymbol{x}_1, \boldsymbol{x}_2; \mathcal{D})e^{f(\boldsymbol{x}_1, \boldsymbol{x}_2; \tilde{\mathcal{D}})}$ for appropriate choices of $f$. Techniques from statistical mechanics, however, allow one to perform heuristic calculations to evaluate such expressions, by alternatively expressing the logarithm of the integral as:

$$\mathbb{E}_{\mathcal{D}, \tilde{\mathcal{D}}} \log \int d\boldsymbol{x}_1 d\boldsymbol{x}_2 w_{\beta_1, \beta_2}(\boldsymbol{x}_1, \boldsymbol{x}_2; \mathcal{D})e^{f(\boldsymbol{x}_1, \boldsymbol{x}_2; \tilde{\mathcal{D}})}$$

$$= \lim_{n \to +0} \frac{\partial}{\partial n} \log\left( \mathbb{E}_{\mathcal{D}, \tilde{\mathcal{D}}}\left[ \int d\boldsymbol{x}_1 d\boldsymbol{x}_2 w_{\beta_1, \beta_2}(\boldsymbol{x}_1, \boldsymbol{x}_2; \mathcal{D})e^{f(\boldsymbol{x}_1, \boldsymbol{x}_2; \tilde{\mathcal{D}})} \right]^n \right), \quad (13)$$

where the second line can be confirmed by a Taylor expansion in $n$. Although the second line of the above expression is not directly computable for arbitrary $n \in \mathbb{R}_{\geq 0}$, it turns out that under plausible assumptions, this can be evaluated for $n \in \mathbb{N}$ in the limit $N \to \infty$. The replica method Mezard et al. (1986); Charbonneau et al. (2023) is based on the assumption that one can analytically continue this formula, defined only for $n \in \mathbb{N}$, to $n \in \mathbb{R}_{\geq 0}$. On this continuation, the limit $n \to +0$ can be taken formally.

In the process of calculating equation 13, a crucial observation is that from Gaussianity of the design matrices $\boldsymbol{A}^{(k)}$, the products between the design matrices and the feature vectors $\boldsymbol{x}_1, \boldsymbol{x}_2$ are Gaussian random variables. This not only simplifies the high-dimensional nature of the expectation over $\mathcal{D}$, but it also reduces the complexity of the analysis to consider only up to the second moments of these Gaussian random variables. These moments are characterized by the inner product of the feature vectors, which happen to concentrate to deterministic values in the limit $N \to \infty$. One is then left with an integral over this finite number of concentrating scalar variables and other auxiliary variables introduced in this process, which can then be

evaluated using the standard saddle-point method, again in the limit $N \to \infty$. This finite set of variables, as well as the saddle-point conditions imposed on them, are given in Definitions 1 and 2 in the following section.

### 3.2 Equations of State and Generalization Error

The precise asymptotic behavior of the generalization error is given by the solution of a set of non-linear equations, which we refer to as the equations of state. They are provided as follows.

**Definition 1** (Equations of state for the First Stage). Define the set of finite scalar variables $\Theta_1 = \{\{m_1^{(k)}, \hat{m}_1^{(k)}\}_{k=0}^K, q_1, \hat{q}_1, \chi_1, \hat{\chi}_1\}$ as the solution to the following set of equations, which we refer to as the equations of state for the first stage:

$$q_1 = \sum_{k=1}^K \pi^{(k)} \mathbb{E}_1 \left[ \left( \mathsf{x}_1^{(k)} \right)^2 \right] + \pi^{(\text{neg.})} \mathbb{E}_1 \left[ \left( \mathsf{x}_1^{(\text{neg.})} \right)^2 \right], \qquad \hat{q}_1 = \hat{m}_1^{(0)} = \frac{\alpha^{(\text{tot})}}{1 + \chi_1},$$

$$m_1^{(k)} = \pi^{(k)} \mathbb{E}_1 \left[ \mathsf{x}_1^{(k)} \mathsf{x}_\star^{(k)} \right], \qquad\qquad\qquad \hat{m}_1^{(k)} = \frac{\alpha^{(k)}}{1 + \chi_1},$$

$$\hat{\chi}_1 = \sum_{k=1}^K \alpha^{(k)} \frac{q_1 - 2(m_1^{(0)} + m_1^{(k)}) + \rho^{(k)}}{(1 + \chi_1)^2}, \tag{14}$$

$$\chi_1 = \sum_{k=1}^K \pi^{(k)} \mathbb{E}_1 \left[ \frac{\partial \mathsf{x}_1^{(k)}}{\partial \sqrt{\hat{\chi}_1} \mathsf{z}_1} \right] + \pi^{(\text{neg.})} \mathbb{E}_1 \left[ \frac{\partial \mathsf{x}_1^{(\text{neg.})}}{\partial \sqrt{\hat{\chi}_1} \mathsf{z}_1} \right],$$

where we have defined $\rho^{(k)} = \pi^{(k)} + \pi^{(0)} + (\sigma^{(k)})^2$, and $\alpha^{(\text{tot})} = \sum_{k=1}^K \alpha^{(k)}$. Here, the random variables $\{\mathsf{x}_1^{(k)}\}_{k=0}^K, \mathsf{x}_1^{(\text{neg.})}$ are the solution to the following random optimization problems dependent on $\{\{\hat{m}_1^{(k)}\}_{k=0}^K, \hat{q}_1, \hat{\chi}_1\}$:

$$\mathsf{x}_1^{(k)} = \arg\min_x \mathsf{E}_1^{(k)}(x), \qquad\qquad \mathsf{E}_1^{(k)}(x) = \frac{\hat{q}_1}{2} x^2 - \left( \sqrt{\hat{\chi}_1} \mathsf{z}_1 + \hat{m}_1^{(k)} \mathsf{x}_\star^{(k)} \right) x + \lambda_1 |x| \tag{15}$$

$$\mathsf{x}_1^{(\text{neg.})} = \arg\min_x \mathsf{E}_1^{(\text{neg.})}(x), \qquad \mathsf{E}_1^{(\text{neg.})}(x) = \frac{\hat{q}_1}{2} x^2 - \sqrt{\hat{\chi}_1} \mathsf{z}_1 x + \lambda_1 |x|, \tag{16}$$

where $\mathsf{z}_1, \{\mathsf{x}_\star^{(k)}\}_{k=0}^K$ are i.i.d. standard Gaussian random variables. Finally, the average $\mathbb{E}_1$ denotes the joint expectation with respect to these random variables.

**Definition 2** (Equations of state of the Second Stage). Let $\Theta_1$ be as defined in 1, i.e. the solution to the equations of state 14. Define the set of finite variables $\Theta_2 = \{\{m_2^{(k)}, \hat{m}_2^{(k)}\}_{k=0}^K, q_2, \hat{q}_2, q_r, \hat{q}_r, \chi_2, \hat{\chi}_2, \chi_r, \hat{\chi}_r\}$ by

the solution to the following set of equations, which we refer to as the equations of state for the first stage:

$$q_2 = \sum_{k=1}^{K} \pi^{(k)} \mathbb{E}_2\left[\left(x_2^{(k)}\right)^2\right] + \pi^{(\text{neg.})} \mathbb{E}_2\left[\left(x_2^{(\text{neg.})}\right)^2\right], \qquad \hat{q}_2 = \frac{\alpha^{(1)}}{1+\chi_2},$$

$$q_r = \sum_{k=1}^{K} \pi^{(k)} \mathbb{E}_2\left[x_2^{(k)} x_1^{(k)}\right] + \pi^{(\text{neg.})} \mathbb{E}_2\left[x_2^{(\text{neg.})} x_1^{(\text{neg.})}\right], \qquad \hat{q}_r = -\frac{\alpha^{(1)} A}{1+\chi_2},$$

$$m_2^{(k)} = \pi^{(k)} \mathbb{E}_2\left[x_2^{(k)} x_\star^{(k)}\right], \qquad \hat{m}_2^{(k)} = \frac{\alpha^{(1)} B}{1+\chi_2},$$

$$\chi_2 = \sum_{k=1}^{K} \pi^{(k)} \mathbb{E}_2\left[\frac{\partial x_2^{(k)}}{\partial \sqrt{\hat{\chi}_2} z_2}\right] + \pi^{(\text{neg.})} \mathbb{E}_2\left[\frac{\partial x_2^{(\text{neg.})}}{\partial \sqrt{\hat{\chi}_2} z_2}\right], \tag{17}$$

$$\hat{\chi}_2 = \alpha^{(1)} \frac{q_2 + A^2 q_1 + 2A q_r + B^2 \rho^{(1)} - 2B(m_2^{(0)} + m_2^{(1)}) - 2AB(m_1^{(0)} + m_1^{(1)})}{(1+\chi_2)^2},$$

$$\chi_r = \sum_{k=1}^{K} \pi^{(k)} \mathbb{E}_2\left[\frac{\partial x_2^{(k)}}{\partial \sqrt{\hat{\chi}_1} z_1}\right] + \pi^{(\text{neg.})} \mathbb{E}_2\left[\frac{\partial x_2^{(\text{neg.})}}{\partial \sqrt{\hat{\chi}_1} z_1}\right],$$

$$\hat{\chi}_r = -\alpha^{(1)} \frac{B\rho^{(1)} - m_2^{(0)} - m_2^{(1)} - A(m_1^{(0)} + m_1^{(1)})}{(1+\chi_1)(1+\chi_2)},$$

where $A = \frac{\kappa - \chi_1}{1+\chi_1}$, $B = 1 - \frac{\chi_r + \kappa \chi_1}{1+\chi_1}$. Here, the random variables $\{x_2^{(k)}\}_{k=0}^{K}, x_2^{(\text{neg.})}$ are the solution to the following random optimization problems dependent on the scalar variables $\{\{\hat{m}_1^{(k)}, \hat{m}_2^{(k)}\}_{k=0}^{K}, \hat{q}_1, \hat{\chi}_1, \hat{q}_2, \hat{\chi}_2, \hat{q}_r\}$:

$$x_2^{(k)} = \arg\min_x E_2^{(k)}(x|x_1^{(k)}), \qquad E_2^{(k)}(x|x_1^{(k)}) = \frac{\hat{q}_2}{2} x^2 - \left(\sqrt{\hat{\chi}_2} z_2 + \hat{m}_2^{(k)} x_\star^{(k)} + \hat{q}_r x_1^{(k)}\right) x + r(x|x_1^{(k)}), \tag{18}$$

$$x_2^{(\text{neg.})} = \arg\min_x E_2^{(\text{neg.})}(x|x_1^{(\text{neg.})}), \qquad E_2^{(\text{neg.})}(x|x_1^{(\text{neg.})}) = \frac{\hat{q}_2}{2} x^2 - \left(\sqrt{\hat{\chi}_2} z_2 + \hat{q}_r x_1^{(\text{neg.})}\right) x + r(x|x_1^{(\text{neg.})}), \tag{19}$$

where $r(x|y) = (\lambda_2 + \Delta\lambda \mathbb{I}[y = 0])|x|$. The random variables $z_1, \{x_\star^{(k)}\}_{k=0}^{K}$ and $\{x_1^{(k)}\}_{k=0}^{K}$ are as predefined in Definition 1, and $z_2$ is a Gaussian random variable with conditional distribution

$$z_2|z_1 \sim \mathcal{N}\left(\frac{\hat{\chi}_r}{\sqrt{\hat{\chi}_1 \hat{\chi}_2}} z_1, 1 - \frac{\hat{\chi}_r^2}{\hat{\chi}_1 \hat{\chi}_2}\right). \tag{20}$$

Finally, the joint expectation with respect to these random variables is denoted as $\mathbb{E}_2$.

Given these equations of state, the generalization error is given from the following claims:

**Claim 1** (Generalization error of the first stage). *Given $\Theta_1$, the expected generalization error $\epsilon^{(\text{1st})}$ converges in the large $N$ limit as*

$$\lim_{N\to\infty} \epsilon^{(\text{1st})} = \sum_{k=1}^{K} \alpha^{(k)} \left[q_1 - 2(m_1^{(0)} + m_1^{(k)}) + \rho^{(k)}\right]. \tag{21}$$

**Claim 2** (Generalization error of the second stage). *Given $\Theta_1$ and $\Theta_2$, the expected generalization error $\epsilon^{(\text{2nd})}$ converges in the large $N$ limit as*

$$\lim_{N\to\infty} \epsilon^{(\text{2nd})} = \alpha^{(1)} \left[q_2 + \kappa^2 q_1 + 2\kappa q_r + \rho^{(1)} - 2(m_2^{(0)} + m_2^{(1)}) - 2\kappa(m_1^{(0)} + m_1^{(1)})\right]. \tag{22}$$

Claims 1 and 2 indicate that for a given set of hyperparameters, the asymptotic generalization error can be computed by solving a finite set of non-linear equations in Definitions 1 and 2. Therefore, one can obtain the optimal choice of hyperparameters minimizing the generalization error via a numerical root-finding and optimization procedure in finite dimension.

### 3.3 Interpretation of $\Theta_1$ and $\Theta_2$

By choosing an appropriate function $f$ in the expression equation 13, we can easily show that the following claims hold.

**Claim 3** (Expected inner product between the regressors). *Recall that $\hat{\boldsymbol{x}}_{\mathcal{I}(k)}^{(\text{1st})}$ and $\hat{\boldsymbol{x}}_{\mathcal{I}(k)}^{(\text{2nd})}$ are subvectors of $\hat{\boldsymbol{x}}^{(\text{1st})}$ and $\hat{\boldsymbol{x}}^{(\text{2nd})}$ with indices corresponding to the unique feature vector for class $k$, $\boldsymbol{x}_\star^{(k)}$. Given $\Theta_1, \Theta_2$, the expected inner product between $\hat{\boldsymbol{x}}_{\mathcal{I}(k)}^{(\text{1st})}$, $\hat{\boldsymbol{x}}_{\mathcal{I}(k)}^{(\text{2nd})}$, and $\boldsymbol{x}_\star^{(k)}$ are given by*

$$\lim_{N \to \infty} \frac{1}{N} \mathbb{E}_{\mathcal{D}, \tilde{\mathcal{D}}} \left[ \hat{\boldsymbol{x}}_{\mathcal{I}(k)}^{(\text{1st})} \cdot \boldsymbol{x}_\star^{(k)} \right] = m_1^{(k)}, \qquad \lim_{N \to \infty} \frac{1}{N} \mathbb{E}_{\mathcal{D}, \tilde{\mathcal{D}}} \left[ \hat{\boldsymbol{x}}_{\mathcal{I}(k)}^{(\text{2nd})} \cdot \boldsymbol{x}_\star^{(k)} \right] = m_2^{(k)}. \tag{23}$$

*Furthermore, we also have that*

$$\lim_{N \to \infty} \frac{1}{N} \mathbb{E}_{\mathcal{D}, \tilde{\mathcal{D}}} \left[ \|\hat{\boldsymbol{x}}^{(\text{1st})}\|_2^2 \right] = q_1, \quad \lim_{N \to \infty} \frac{1}{N} \mathbb{E}_{\mathcal{D}, \tilde{\mathcal{D}}} \left[ \|\hat{\boldsymbol{x}}^{(\text{2nd})}\|_2^2 \right] = q_2, \quad \lim_{N \to \infty} \frac{1}{N} \mathbb{E}_{\mathcal{D}, \tilde{\mathcal{D}}} \left[ \hat{\boldsymbol{x}}^{(\text{1st})} \cdot \hat{\boldsymbol{x}}^{(\text{2nd})} \right] = q_r, \tag{24}$$

*that is, $q_1, q_2$ denote the norm of the first and second stage regressors respectively, while $q_r$ denotes the inner product between the first and second stage regressors.*

Claim 3 not only clarifies the interpretation of the some of the parameters in $\Theta_1$ and $\Theta_2$, but it also provides an alternative derivation of the expressions in Claims 1 and 2. For instance, $\epsilon^{(\text{1st})}$ can be expressed alternatively as

$$\epsilon^{(\text{1st})} = \lim_{N \to \infty} \frac{1}{N} \sum_{k=1}^{K} \mathbb{E}_{\mathcal{D}, \tilde{\mathcal{D}}} \left[ \left\| \tilde{\boldsymbol{A}}_{\mathcal{I}(0)}^{(k)} \big( \boldsymbol{x}_\star^{(0)} - \hat{\boldsymbol{x}}_{\mathcal{I}(0)}^{(\text{1st})} \big) + \tilde{\boldsymbol{A}}_{\mathcal{I}(k)}^{(k)} \big( \boldsymbol{x}_\star^{(0)} - \hat{\boldsymbol{x}}_{\mathcal{I}(k)}^{(\text{1st})} \big) - \sum_{k' \neq 0, k} \tilde{\boldsymbol{A}}_{\mathcal{I}(k')}^{(k)} \hat{\boldsymbol{x}}_{\mathcal{I}(k')}^{(\text{1st})} + \boldsymbol{e}^{(k)} \big) \right\|_2^2 \right]$$

$$= \lim_{N \to \infty} \sum_{k=1}^{K} \frac{M_k}{N} \mathbb{E}_{\mathcal{D}} \left[ \|\boldsymbol{x}_\star^{(k)} - \hat{\boldsymbol{x}}_{\mathcal{I}(0)}^{(\text{1st})}\|_2^2 + \|\boldsymbol{x}_\star^{(0)} - \hat{\boldsymbol{x}}_{\mathcal{I}(k)}^{(\text{1st})}\|_2^2 + \sum_{k' \neq 0, k} \|\hat{\boldsymbol{x}}_{\mathcal{I}(k')}^{(\text{1st})}\|_2^2 + (\sigma^{(k)})^2 \right], \tag{25}$$

which results in the expression in Claim 1 by using equations 23 and 24. Similar calculations can be made for the second stage generalization error $\epsilon^{(\text{2nd})}$, in Claim 2.

## 4 Comparison with synthetic finite-size data simulations

To verify our theoretical results, we conduct a series of synthetic numerical experiments to compare the results obtained from Claim 2 and the empirical results for the generalization error obtained from datasets of finite-size systems.

The synthetic data is specified as follows. We consider the case of $K = 2$ classes, with $\alpha^{(1)} = 0.2, 0.4$, or $0.6$, and $\alpha^{(2)} = 0.8$. Two types of settings for the common and unique feature vectors are considered; one with $\pi^{(0)} = 0.10, \pi^{(1)} = \pi^{(2)} = 0.09$, and $\pi^{(0)} = 0.15, \pi^{(1)} = \pi^{(2)} = 0.04$. These two settings are chosen to represent the cases where the common feature vector is relatively small compared to the unique feature vectors, and the case where the common feature vector is relatively large, but with the total number of non-zero features underlying each class being the same ($0.19N$). Finally, the noise level of each dataset is set to $\sigma^{(1)} = \sigma^{(2)} = 0.1$. The numerical simulations are performed on 64 random realizations of data with size $N = 16,000$. The second stage generalization error is evaluated by further generating 256 random realizations of $(\tilde{\boldsymbol{A}}^{(1)}, \tilde{\boldsymbol{y}}^{(1)})$ for each dataset and calculating the average empirical generalization error over these realizations. The hyperparameters $\lambda_1$ and $\lambda_2$ are chosen such that they minimize the generalization error given in Claims 1 and 2 for each stage respectively. The finite size simulations are also performed using the same values of $\lambda_1$ and $\lambda_2$ used in the theory. In figure 1, we plot the generalization error of the second stage as a function of $\Delta\lambda$ for $\kappa = 0.0, 0.2, 0.5$, and $1.0$. As evident from the plot, the generalization error is in fairly good agreement with the theoretical predictions.

**Case with scarce target data.** Let us look closely at the left column of figure 1 where the target sample size is relatively scarce ($\alpha^{(1)} = 0.2$). For both settings of $(\pi^{(0)}, \pi^{(1)}, \pi^{(2)})$, the generalization error

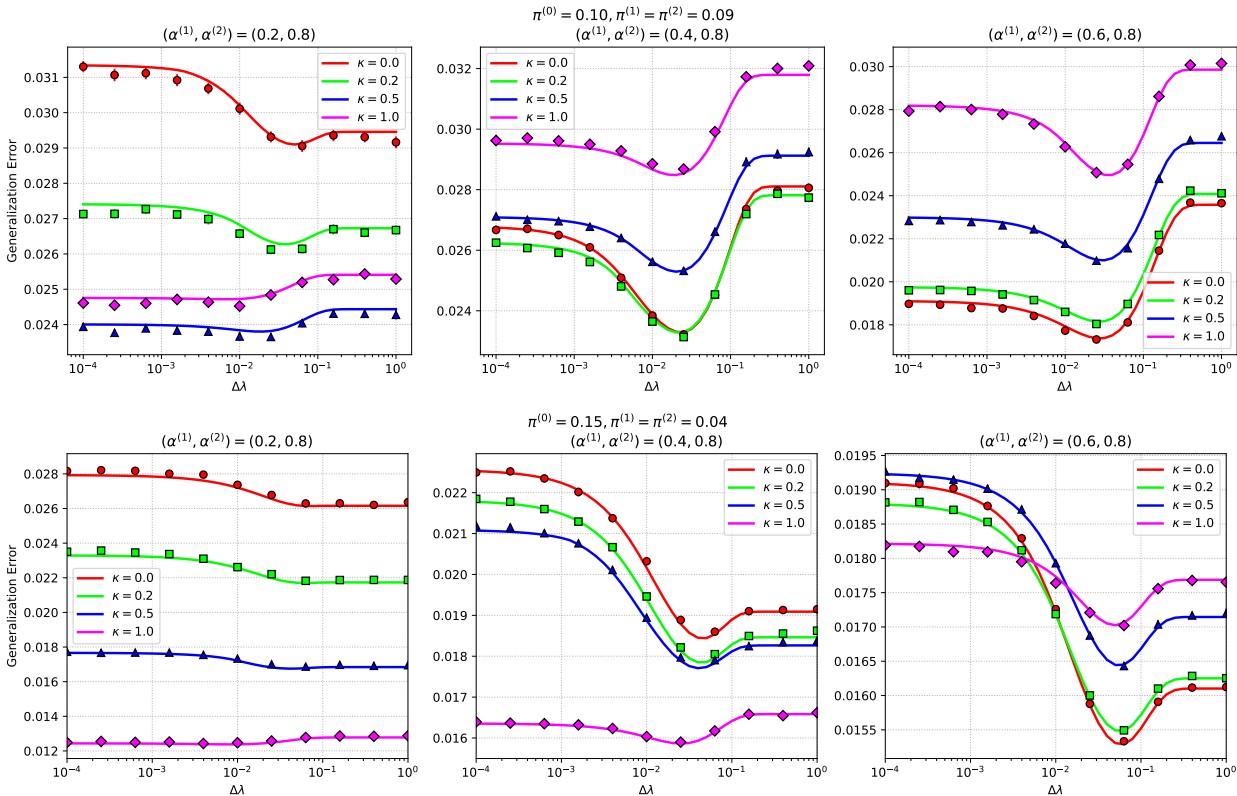

Figure 1: Comparison of generalization error obtained from Claim 2 (solid line) with finite size simulations (markers). Error bars represent standard error obtained from 64 realizations of data.

is insensitive to the choice of $\Delta\lambda$ when $\kappa$ is tuned properly. Another important observation is that Trans-Lasso's hyperparameter choice ($\kappa = 1, \Delta\lambda = 0$) seems comparable with the optimal choice regarding the generalization error. The same trend is also seen in the case where $\alpha^{(1)} = 0.4$ and $(\pi^{(0)}, \pi^{(1)}, \pi^{(2)}) = (0.15, 0.04, 0.04)$ (bottom center subfigure of figure 1).

**Case with abundant target data.** Next, we turn our attention to the right column of figure 1 where the target sample size is relatively abundant ($\alpha^{(1)} = 0.6$). In this case, the generalization error is sensitive to the choice of $\Delta\lambda$ but is insensitive to the choice of $\kappa$ around its optimal value ($\kappa = 0.0$), for both settings of $(\pi^{(0)}, \pi^{(1)}, \pi^{(2)})$. This indicates that under abundant target data, transferring the knowledge with respect to the support of the pretrained feature vector $\hat{\boldsymbol{x}}^{(\mathsf{1st})}$ is more crucial than the actual value of $\hat{\boldsymbol{x}}^{(\mathsf{1st})}$ itself. The same trend can be seen for the case when $\alpha^{(1)} = 0.4$ and $(\pi^{(0)}, \pi^{(1)}, \pi^{(2)}) = (0.15, 0.04, 0.04)$ (top center subfigure of figure 1).

## 5 Hyperparameter Selection Strategy

The experiments from the last section indicate not only the accuracy of our theoretical predictions but also the practical implications for hyperparameter selection. The observations made from the two cases suggest that there are roughly two behaviors for the optimal choice of hyperparameters: One where the choice of $\kappa$ is more crucial, and the other where the choice of $\Delta\lambda$ is more important. This observation suggests a simple strategy for hyperparameter selection, where one fixes $\Delta\lambda = 0$ for the former case and $\kappa = 0$ for the latter case. To further investigate the validity of this strategy, we conduct a comparative analysis of the generalization error among the following four strategies:

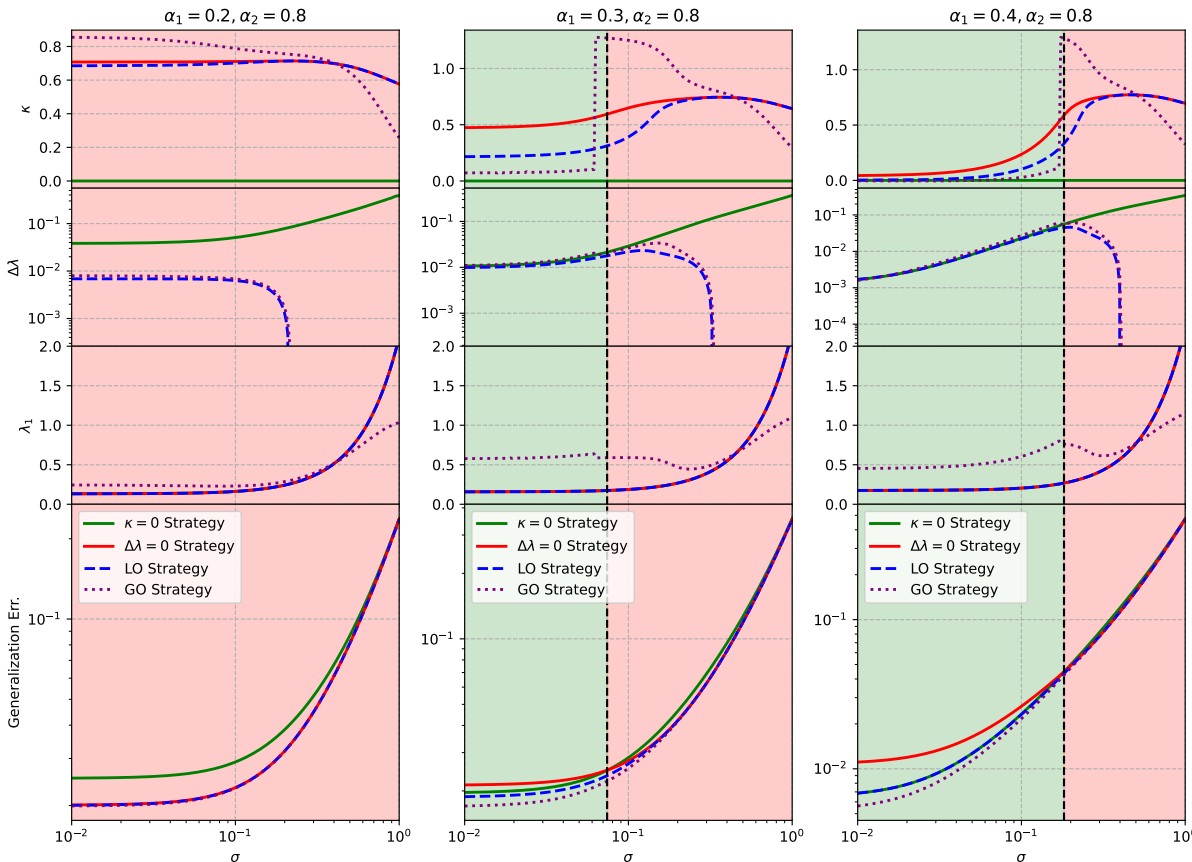

Figure 2: Generalization error and optimal hyperparameters for each strategy as a function of $\sigma$ for $(\alpha_1, \alpha_2) = (0.2, 0.8), (0.3, 0.8), (0.4, 0.8)$. The region shaded in red indicates the range of $\sigma$ where the $\Delta\lambda = 0$ strategy outperforms the $\kappa = 0$ strategy, while the region shaded in green indicates the opposite.

- $\kappa = 0$ **strategy.** Here, we fix $\kappa$ to zero, while letting $\Delta\lambda$ to be a tunable parameter. This reflects on the insight obtained in subsection 4 for the abundant target data case. The regularization parameters $\lambda_1$ and $\lambda_2$ are both chosen such that they minimize the generalization error in the first and second stages, respectively.

- $\Delta\lambda = 0$ **strategy.** Here, we fix $\Delta\lambda$ to zero, while letting $\kappa$ to be a tunable parameter. This reflects the insight obtained in subsection 4 for the scarce target data case. The regularization parameters $\lambda_1$ and $\lambda_2$ are both chosen in the same way as the $\kappa = 0$ strategy.

- **Locally Optimal (LO) strategy.** Here, both $\kappa$ and $\Delta\lambda$ are tunable parameters. The regularization parameters $\lambda_1$ and $\lambda_2$ are both chosen in the same way as the $\kappa = 0$ strategy.

- **Globally Optimal (GO) strategy.** Here, all four parameters $(\lambda_1, \lambda_2, \kappa, \Delta\lambda)$ are tuned such that the second stage generalization error is minimized.

The last strategy requires retraining the first stage estimator $\hat{\boldsymbol{x}}^{(\mathsf{1st})}$ for each choice of $(\lambda_1, \lambda_2, \kappa, \Delta\lambda)$, which is computationally demanding. Even the LO strategy can be problematic when the dataset size is large. Basically, these two strategies are regarded as a benchmark for the other two strategies: $\kappa = 0$ and $\Delta\lambda = 0$ strategies. Below, we compare these four strategies in terms of the second-stage generalization error computed by our theoretical analysis. Specifically, the case $(\pi^{(0)}, \pi^{(1)}, \pi^{(2)}) = (0.10, 0.09, 0.09)$ is investigated. In figure 2, we plot the second stage generalization error, as well as the optimal choice of hyperparameters for each strategy as a function of $\sigma$, for $(\alpha_1, \alpha_2) = (0.2, 0.8), (0.3, 0.8)$, and $(0.4, 0.8)$. Note that the shaded green

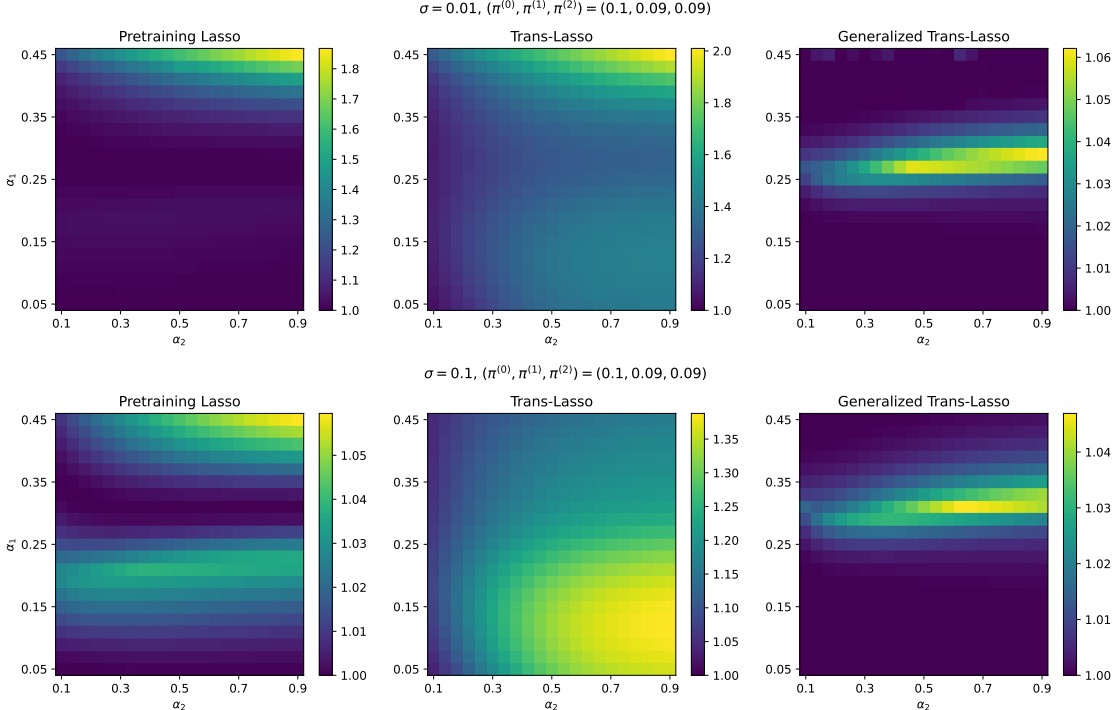

Figure 3: Ratios $\min\{\epsilon_{\Delta\lambda=0}, \epsilon_{\kappa=0}\}/\epsilon_{\mathrm{LO}}, \epsilon_{\mathrm{Pretrain}}/\epsilon_{\mathrm{LO}}$ and $\epsilon_{\mathrm{Trans}}/\epsilon_{\mathrm{LO}}$ under setting $(\pi^{(0)}, \pi^{(1)}, \pi^{(2)}) = (0.1, 0.09, 0.09)$ and noise level $\sigma = 0.01$ (top figure) and $0.1$ (bottom figure) for various values of $(\alpha^{(1)}, \alpha^{(2)})$. Note that each heatmap has its own color scale bar; see Appendix B for a direct comparison with shared color scale bars between the $\kappa = 0$ or $\Delta\lambda = 0$ strategy with Pretraining-Lasso and Trans-Lasso.

region indicates the range of $\sigma$ where the $\kappa = 0$ strategy outperforms the $\Delta\lambda = 0$ strategy, while the shaded red region indicates the opposite.

Let us first examine the behavior of optimal $\Delta\lambda$ across different strategies. With the exception of the case $(\alpha_1, \alpha_2) = (0.2, 0.8)$, the optimal $\Delta\lambda$ values for each strategy (excluding the $\Delta\lambda = 0$ strategy) are approximately equal in the green region. Notably, for $(\alpha^{(1)}, \alpha^{(2)}) = (0.4, 0.8)$, the $\kappa = 0$ strategy demonstrates generalization performance comparable to the LO strategy in this region, while the $\Delta\lambda = 0$ strategy exhibits significantly lower performance. Above a certain noise threshold, the optimal value of $\Delta\lambda$ for both the LO and GO strategies becomes strictly zero. At this point, the LO strategy and the $\Delta\lambda = 0$ strategy coincide, indicating that the $\Delta\lambda = 0$ strategy can potentially be equivalent to the LO strategy.

Next, let us see the behavior of the optimal $\kappa$ in each strategy. In the green region, where the $\kappa = 0$ strategy outperforms the $\Delta\lambda = 0$ strategy, both the LO and GO strategies actually exhibit finite values of $\kappa$. This means that the $\kappa = 0$ strategy is suboptimal. However, it still exhibits a comparable performance with the LO strategy and gives a significant improvement from the $\Delta\lambda = 0$ strategy. This underscores the importance of solely transferring support information; transferring the first stage feature vector itself can be less beneficial. This may be due to the crosstalk noise induced by different datasets: Using the regression results on other datasets than the target can induce noises on the feature vector components specific to the target. Consequently, it may be preferable to transfer more robust information, in this case the support configuration, rather than the noisy vector itself obtained in the first stage.

The above observations indicate that the performance closely comparable to the LO strategy can be achieved across a wide range of noise levels by simply considering either the $\kappa = 0$ or $\Delta\lambda = 0$ strategy; the $\kappa = 0$ strategy is preferred in low noise scenarios while the $\Delta\lambda = 0$ strategy is better in high noise scenarios. For practical purposes, these two strategies should be examined and the best hyperparameters should be chosen based on comparison. This is the main message for practitioners in this paper.

| Method | Drama | Comedy | Horror |
|---|---|---|---|
| Gen. Trans-Lasso, $\Delta\lambda = 0$ | $\mathbf{5.58 \pm 0.07}$ | $\mathbf{5.78 \pm 0.10}$ | $\mathbf{5.07 \pm 0.11}$ |
| Gen. Trans-Lasso, $\kappa = 0$ | $5.67 \pm 0.08$ | $5.90 \pm 0.10$ | $5.48 \pm 0.13$ |
| Gen. Trans-Lasso, LO | $\mathbf{5.58 \pm 0.07}$ | $\mathbf{5.78 \pm 0.10}$ | $5.19 \pm 0.12$ |
| Pretraining Lasso | $5.63 \pm 0.07$ | $\mathbf{5.78 \pm 0.10}$ | $5.19 \pm 0.12$ |
| Trans-Lasso | $5.71 \pm 0.08$ | $\mathbf{5.78 \pm 0.10}$ | $5.17 \pm 0.12$ |

Table 2: Comparison of test errors for various Lasso-based methods and hyperparameter strategies applied to the IMDb dataset. Error estimation on the test error is done by performing jackknife resampling on the test dataset.

**Implications on the original Trans-Lasso.** The $\Delta\lambda = 0$ strategy provides a lower bound for the generalization error of the Trans-Lasso, as the latter constrains $\kappa$ to unity instead of treating it as a tunable parameter. Our results demonstrate that the Trans-Lasso is suboptimal compared to strategies that incorporate support information in the second stage, particularly in the green region of figure 2. Notably, for $(\alpha_1, \alpha_2) = (0.4, 0.8)$ and $\sigma = 0.01$, the $\Delta\lambda = 0$ strategy reduces the generalization error by over 30% compared to the Trans-Lasso.

**Ubiquity of the simplified hyperparameter selection.** To further confirm the efficacy of the two simple strategies, we investigate the difference in generalization error between the LO strategy and $\kappa = 0$ or $\Delta\lambda = 0$ strategy. More specifically, let $\epsilon_{\text{LO}}$, $\epsilon_{\kappa=0}$ and $\epsilon_{\Delta\lambda=0}$ be the generalization error of the corresponding strategies. Figure 3 shows the minimum ratio $\min\{\epsilon_{\Delta\lambda=0}, \epsilon_{\kappa=0}\}/\epsilon_{\text{LO}}$ for various values of $(\alpha_1, \alpha_2)$. For comparison, the same values are calculated for the Pretraining Lasso and Trans-Lasso, i.e. $\epsilon_{\text{Pretrain}}/\epsilon_{\text{LO}}$ and $\epsilon_{\text{Trans}}/\epsilon_{\text{LO}}$, with $\epsilon_{\text{Pretrain}}$ and $\epsilon_{\text{Trans}}$ being the generalization error with the parameters chosen optimally (note that $\lambda_1$ and $\lambda_2$ are chosen in the same way as the LO strategy). The results indicate that, across a broad range of parameters, at least one of the simplified strategies (either $\Delta\lambda = 0$ or $\kappa = 0$) achieves a generalization error within 10% of that obtained by the more complex LO strategy. Notably, substantial relative differences are observed only within a narrow region of the parameter space $(\alpha^{(1)}, \alpha^{(2)})$, highlighting the effectiveness of the simplified approach across a wide range of problem settings. We also report that for the case $\sigma = 0.5$, the $\Delta\lambda = 0$ strategy always coincided with the LO strategy in the region $(\alpha^{(1)}, \alpha^{(2)}) \in [0.05, 0.45] \times [0.1, 0.9]$. Results consistent with the above observations were also seen for other settings of $(\pi^{(0)}, \pi^{(1)}, \pi^{(2)})$; see Appendix B.

**The role of $\lambda_1$.** For low noise scenarios, the GO strategy tends to prefer larger $\lambda_1$ in the first stage compared to the other strategies. This observation may be beneficial when the noise level of each dataset is known a priori: We may tune the $\lambda_1$ value to a larger value than the one obtained from the LO or its proxy strategies ($\kappa = 0$ or $\Delta\lambda = 0$ strategies). This offers an opportunity to enhance our prediction beyond the LO strategy while avoiding the complicated retraining process necessary for the GO strategy.

## 6 Real Data Application

### 6.1 Application to the IMDb dataset

To verify whether the above insights can be generalized to real-world scenarios, we here conduct experiments on the IMDb dataset Maas et al. (2011). The IMDb dataset comprises a pair of 25,000 user-generated movie reviews from the Internet Movie Database, each being devoted to training and test datasets. The dataset is evenly split into highly positive or negative reviews, with negative labels assigned to ratings given 4 or lower out of 10, and positive labels to ratings of 7 or higher. Movies labeled with only one of the three most common genres in the dataset, drama, comedy, and horror, are considered. By representing the reviews as binary feature vectors using bag-of-words while ignoring words with less than 5 occurrences, we are left with 27743 features in total, with 8286 drama, 5027 comedy, and 3073 horror movies in the training dataset, and 9937 drama, 4774 comedy, and 3398 horror movies in the test dataset. The datasets can be acquired from Supplementary Materials of Gross & Tibshirani (2016).

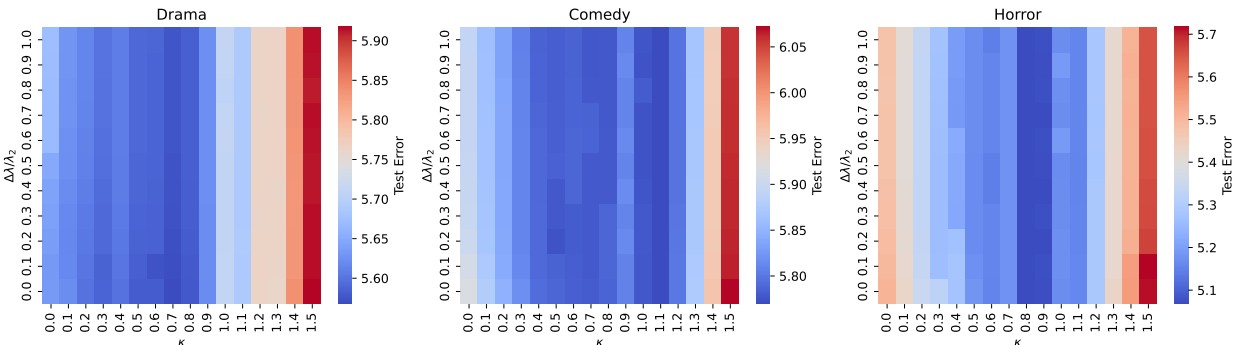

Figure 4: Second stage test error for each IMDB genre plotted against varying values of hyperparameters $\kappa$ and $\Delta\lambda/\lambda_2$. The plots demonstrate that the effect of $\Delta\lambda$ on generalization performance is tiny across genres.

For sake of completeness, we calculate the test error for $(\kappa, \Delta\lambda/\lambda_2) \in \{0.0, 0.1, 0.2, \ldots, 1.5\} \times \{0.0, 0.1, 0.2, \ldots, 1.0\}$ to see if our insights are qualitatively consistent with experiments on real data. For each pair of $(\kappa, \Delta\lambda/\lambda_2)$, $\lambda_1$ and $\lambda_2$ are determined by a 10-fold cross-validation procedure, where the training dataset is further split into a learning dataset and a validation dataset. Since $\lambda_1$ (and consequently $\hat{x}^{(1st)}$) must be selected oblivious to the validation dataset, we perform Approximate Leave-One-Out Cross-Validation (Obuchi & Kabashima, 2016; Rad & Maleki, 2020; Stephenson & Broderick, 2020) on the learning dataset, which allows one to estimate the leave-one-out cross-validation error deterministically without actually performing the computationally expensive procedure. For the sake of comparison, we also compare with the performance of Pretraining Lasso and Trans-Lasso, with $\lambda_1$ and $\lambda_2$ chosen using the same method as above.

In Figure 4, we present the second stage test error for each genre across various values of $(\kappa, \Delta\lambda/\lambda_2)$. The plot reveals that $\Delta\lambda$ is essentially a redundant hyperparameter having almost no influence on generalization performance. This observation strongly supports the efficacy of the $\Delta\lambda = 0$ strategy in the present dataset. We compare the test errors obtained from three approaches within our generalized Trans-Lasso framework: The $\Delta\lambda = 0$ strategy, the $\kappa = 0$ strategy, and LO strategy. Additionally, we include test errors from Pretraining Lasso and the original Trans-Lasso for comparison, whose results are summarized in Table 2. Recall that Trans-Lasso fixes $\kappa$ to unity, while Pretraining Lasso employs $(\kappa, \Delta\lambda/\lambda_2) = (1-s, (1-s)/s)$ for hyperparameter $s \in [0, 1]$. In the experiments, we choose $s$ from $\{0.0, 0.1, 0.2, \ldots, 1.0\}$ via cross-validation. Our analysis demonstrates that the hyperparameter choices made by both the original Trans-Lasso and Pretraining Lasso are suboptimal compared to the $\Delta\lambda = 0$ strategy of our generalized Trans-Lasso algorithm. Note that the LO strategy can potentially select hyperparameters exhibiting higher test error compared to the $\Delta\lambda = 0$ strategy (as seen in the horror genre), as they are chosen via cross-validation and the true generalization error is not directly minimized.

## 6.2 Application to Compressed Imaging task on the MNIST dataset

The performance of the Generalized Trans-Lasso algorithm is also evaluated on a compressed imaging task (Donoho, 2006; Takhar et al., 2006; Lustig et al., 2007; Romberg, 2008; Candès & Wakin, 2008) on the MNIST dataset (Deng, 2012). The MNIST dataset consists of grayscale images of handwritten digits with $28 \times 28$ pixels, totaling 784 dimensions. Given noisy, linear measurements of the first set of 3 handwritten images, "1", "7", and "9" appearing in the MNIST dataset, the task is to refine the recovery performance of each image in the wavelet basis by transfer learning techniques. We consider the Gaussian measurement setup for vectorized images $\boldsymbol{x}_{"n"} \in \mathbb{R}^{784}$ $(n = 1, 7, 9)$, where measurements are given by $\boldsymbol{y}_{"n"} = \boldsymbol{H}_{"n"}\boldsymbol{x}_{"n"} + \boldsymbol{e}_{"n"} \in \mathbb{R}^{M_{"n"}}$. The matrices $\boldsymbol{H}_{"n"}$ consist of i.i.d. standard Gaussian elements, while $\boldsymbol{e}_{"n"}$ represents additive Gaussian noise with i.i.d. centered elements calibrated to achieve uniform signal-to-noise ratio across measurements. The measurement dimensions are set to $M_{"1"} = 200$, $M_{"7"} = 400$, and $M_{"9"} = 600$. The image recovery task is to

reconstruct $\boldsymbol{x}_{«n»}$ under the assumption that its wavelet transform $\boldsymbol{\theta}_{«n»} = \mathcal{W}^{-1}\boldsymbol{x}_{«n»}$ is sparse, where $\mathcal{W}$ is the 2-d wavelet transformation matrix. Therefore, all algorithms, generalized Trans-Lasso, Pretraining Lasso, and original Trans-Lasso, are targeted to estimate the coefficients of each image in the wavelet basis, $\boldsymbol{\theta}_{«n»}$, given observations $\boldsymbol{y}_{«n»}$ and covariate matrix $\boldsymbol{A}_{«n»} = \boldsymbol{H}_{«n»}\mathcal{W}^{-1}$. The test error is evaluated by generating another random instance $\tilde{\boldsymbol{y}}_{«n»}$ with the same statistical profile and dimension as $\boldsymbol{y}_{«n»}$, and using this as the test dataset.

We calculate the test error on a $(\kappa, \Delta\lambda/\lambda_2)$ grid, where $\kappa = \{0.0, 0.1, 0.2, \cdots, 1.5\}$, and $\Delta\lambda/\lambda_2$ is taken from 21 logarithmically equidistanced points between $10^{-2}$ and 10. Other procedures are equivalent to those of the IMDb experiment.

In Figure 5, we present the second stage test error for different handwritten digits across various values of $(\kappa, \Delta\lambda)$ under different SNR conditions. Consistent with our findings from synthetic experiments, the support information plays a crucial role in improving generalization performance, particularly at high SNR values. The test errors, with the hyperparameters chosen via cross-validation error minimization, are summarized in Table 3. For high signal quality (SNR = 20), the $\kappa = 0$ strategy demonstrates superior performance. However, as the SNR decreases, the performance gap between different strategies diminishes, with differences in test error becoming statistically insignificant. This empirical observation aligns with our theoretical analysis (Figure 2), where the generalization error curves for the LO, $\Delta\lambda = 0$, and $\kappa = 0$ strategies collapse into one curve at high noise levels.

| | SNR = 20 | | |
|---|---|---|---|
| Method | $n = 1$ | $n = 7$ | $n = 9$ |
| Gen. Trans-Lasso, LO | **0.0721 ± 0.0069** | **0.0361 ± 0.0029** | **0.0139 ± 0.0009** |
| Gen. Trans-Lasso, $\kappa = 0$ | **0.0721 ± 0.0069** | 0.0367 ± 0.0029 | **0.0139 ± 0.0009** |
| Gen. Trans-Lasso, $\Delta\lambda = 0$ | 0.0816 ± 0.0076 | 0.0404 ± 0.0030 | 0.0144 ± 0.0009 |
| Pretraining Lasso | 0.0847 ± 0.0078 | 0.0381 ± 0.0029 | 0.0140 ± 0.0009 |
| Trans-Lasso | 0.1333 ± 0.0122 | 0.0499 ± 0.0037 | 0.0171 ± 0.0010 |

| | SNR = 5 | | |
|---|---|---|---|
| Method | $n = 1$ | $n = 7$ | $n = 9$ |
| Gen. Trans-Lasso, LO | **0.163 ± 0.013** | **0.103 ± 0.008** | 0.0453 ± 0.0028 |
| Gen. Trans-Lasso, $\kappa = 0$ | **0.163 ± 0.013** | **0.103 ± 0.008** | **0.0450 ± 0.0028** |
| Gen. Trans-Lasso, $\Delta\lambda = 0$ | 0.169 ± 0.014 | 0.106 ± 0.008 | 0.0460 ± 0.0028 |
| Pretraining Lasso | 0.165 ± 0.014 | **0.103 ± 0.008** | 0.0454 ± 0.0028 |
| Trans-Lasso | 0.215 ± 0.020 | 0.117 ± 0.009 | 0.0504 ± 0.0029 |

| | SNR = 2 | | |
|---|---|---|---|
| Method | $n = 1$ | $n = 7$ | $n = 9$ |
| Gen. Trans-Lasso, LO | 0.292 ± 0.024 | **0.196 ± 0.015** | **0.099 ± 0.006** |
| Gen. Trans-Lasso, $\kappa = 0$ | **0.290 ± 0.025** | **0.196 ± 0.015** | 0.101 ± 0.006 |
| Gen. Trans-Lasso, $\Delta\lambda = 0$ | 0.303 ± 0.026 | 0.203 ± 0.015 | 0.101 ± 0.006 |
| Pretraining Lasso | 0.292 ± 0.025 | **0.196 ± 0.015** | **0.099 ± 0.006** |
| Trans-Lasso | 0.303 ± 0.027 | 0.208 ± 0.016 | 0.100 ± 0.006 |

Table 3: Performance comparison of different methods across the three handwritten digits for SNR = 20, 5, and 2. Error estimation on the test error is done by performing jackknife resampling on the test dataset.

## 7 Conclusion

In this work, we have conducted a sharp asymptotic analysis of the generalized Trans-Lasso algorithm, precisely characterizing the effect of hyperparameters on its generalization performance. Our theoretical calculations reveal that near-optimal generalization in this transfer learning algorithm can be achieved by focusing on just one of two modes of knowledge transfer from the source dataset. The first mode transfers

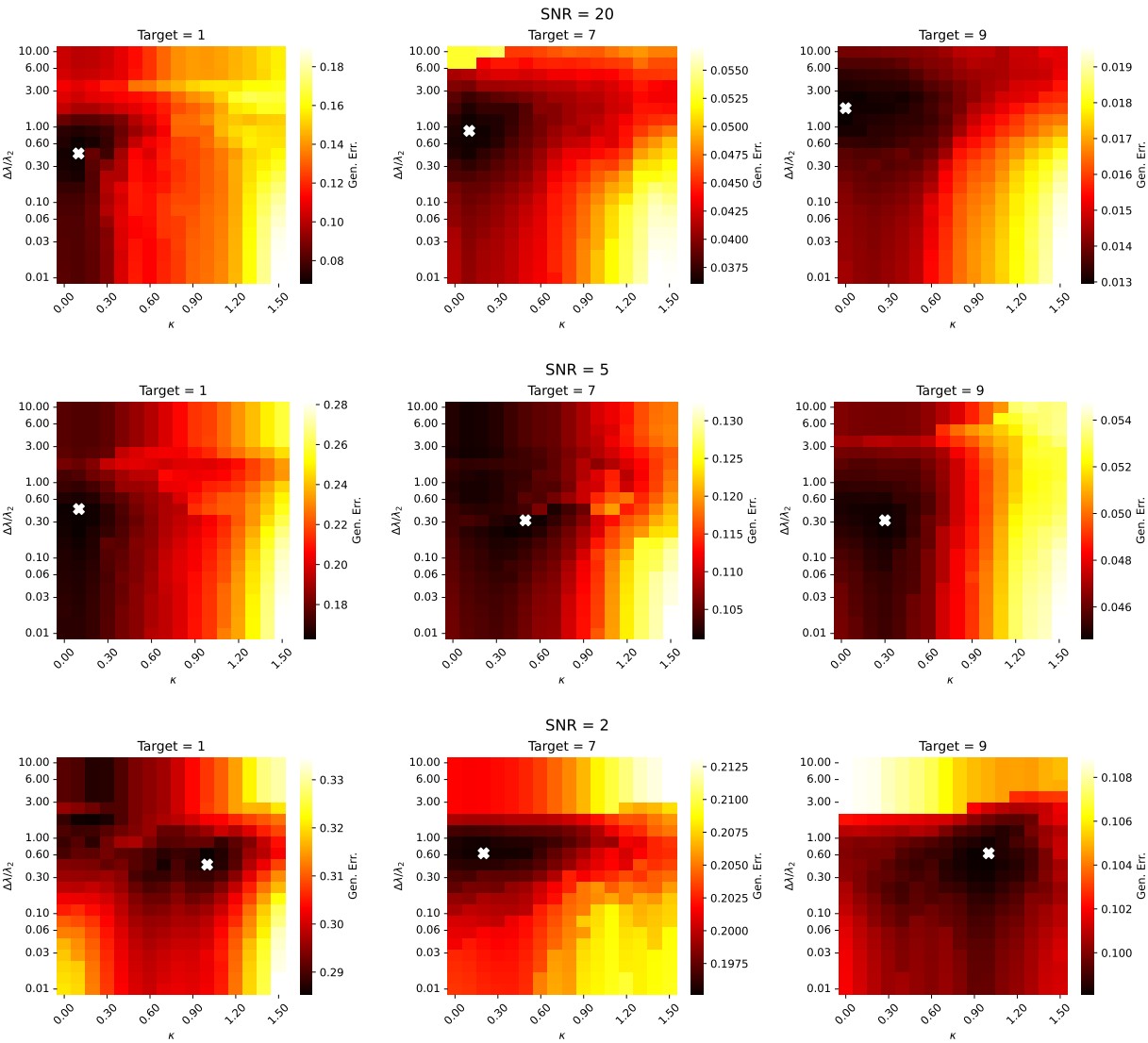

Figure 5: Second stage test error for each MNIST image for SNR = 20 (top), 5 (middle) ,and 2 (bottom), plotted against varying values of hyperparameters $\kappa$ and $\Delta\lambda/\lambda_2$. White markers are placed at the minimizer of the test error for sake of visualization. The plots demonstrate that the effect of $\Delta\lambda$ on generalization performance is nontrivial.

only the support information of the features obtained from all source data, while the second transfers the actual configuration of the feature vector obtained from all sources. Experiments using the IMDb and MNIST dataset confirm that this simple hyperparameter strategy is effective, potentially outperforming conventional Lasso-type transfer algorithms.

## Acknowledgements

This work was supported by JSPS KAKENHI Grant Nos. 22KJ1074 (KO), 22K12179 (TO) and Grant-in-Aid for Transformative Research Areas (A), "Foundation of Machine Learning Physics" (22H05117) (TO).

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

## A  Detailed derivation of Asymptotic analysis

Here, we provide a detailed derivation for the asymptotic formulae for the generalization errors $\epsilon^{\text{(1st)}}$ and $\epsilon^{\text{(2nd)}}$. Recall that to calculate the generalization error, an appropriate choice of $f$ in equation 13 is

$$f(\boldsymbol{x}_1, \boldsymbol{x}_2; \tilde{\mathcal{D}}) = t_1 \sum_{k=1}^{K} \|\tilde{\boldsymbol{y}}^{(k)} - \tilde{\boldsymbol{A}}^{(k)} \boldsymbol{x}_1\|_2^2 + t_2 \|\tilde{\boldsymbol{y}}^{(1)} - \tilde{\boldsymbol{A}}^{(1)}(\kappa \boldsymbol{x}_1 + \boldsymbol{x}_2)\|_2^2. \tag{26}$$

Therefore, the objective of our analysis is to calculate the data average of the $n-$th power of

$$\Phi(\mathcal{D}, \tilde{\mathcal{D}}) := \int d\boldsymbol{x}_1 d\boldsymbol{x}_2 e^{-\beta_1 \mathcal{L}^{\text{(1st)}}(\boldsymbol{x}_1; \mathcal{D}) - \beta_2 \mathcal{L}^{\text{(2nd)}}(\boldsymbol{x}_2; \boldsymbol{x}_1, \mathcal{D}^{(1)}) + t_1 \sum_{k=1}^{K} \|\tilde{\boldsymbol{y}}^{(k)} - \tilde{\boldsymbol{A}}^{(k)} \boldsymbol{x}_1\|_2^2 + t_2 \|\tilde{\boldsymbol{y}}^{(1)} - \tilde{\boldsymbol{A}}^{(1)}(\kappa \boldsymbol{x}_1 + \boldsymbol{x}_2)\|_2^2}, \tag{27}$$

in the successive limit $\lim_{\beta_2 \to \infty} \lim_{\beta_1 \to \infty}$. For $n \in \mathbb{N}$, one can express this as

$$\mathbb{E}_{\mathcal{D}, \tilde{\mathcal{D}}}\Big[\Phi^n(\mathcal{D}, \tilde{\mathcal{D}})\Big] = \int \left( \prod_{a=1}^{n} d\boldsymbol{x}_{1,a} d\boldsymbol{x}_{2,a} \right) \exp \left[ -\sum_{a=1}^{n} \Big( \beta_1 \lambda_1 \|\boldsymbol{x}_{1,a}\|_1 + \beta_2 r(\boldsymbol{x}_{2,a}|\boldsymbol{x}_{1,a}) \Big) \right] \mathcal{L},$$

$$\mathcal{L} = \mathbb{E}_{\mathcal{D}, \tilde{\mathcal{D}}} \prod_{\mu=1}^{M} \prod_{a=1}^{n} \exp \Bigg[ -\frac{\beta_1}{2} \sum_{k=1}^{K} \big(y_\mu^{(k)} - \boldsymbol{A}_{\mu,:}^{(k)} \cdot \boldsymbol{x}_{1,a}\big)^2 - t_1 \sum_{k=1}^{K} \big(\tilde{y}_\mu^{(k)} - \tilde{\boldsymbol{A}}_{\mu,:}^{(k)} \cdot \boldsymbol{x}_{1,a}\big)^2 \tag{28}$$

$$- \frac{\beta_2}{2} \big(y_\mu^{(1)} - \boldsymbol{A}_{\mu,:}^{(1)} \cdot (\kappa \boldsymbol{x}_{1,a} + \boldsymbol{x}_{2,a})\big)^2 - t_2 \big(\tilde{y}_\mu^{(1)} - \tilde{\boldsymbol{A}}_{\mu,:}^{(1)} \cdot (\kappa \boldsymbol{x}_{1,a} + \boldsymbol{x}_{2,a})\big)^2 \Bigg].$$

Conditioned on $\{\boldsymbol{x}_{1,a}, \boldsymbol{x}_{2,a}\}_{a=1}^{n}$, the distribution over $\{\boldsymbol{A}_{\mu,:}^{(k)} \cdot \boldsymbol{x}_{1,a}, \tilde{\boldsymbol{A}}_{\mu,:}^{(k)} \cdot \boldsymbol{x}_{1,a}, \boldsymbol{A}_{\mu,:}^{(1)} \cdot \boldsymbol{x}_{2,a}, \tilde{\boldsymbol{A}}_{\mu,:}^{(1)} \cdot \boldsymbol{x}_{2,a}, y_\mu, \tilde{y}_\mu\}_{\mu=1}^{M}$ is i.i.d. with respect to $\mu$, whose profile is identical to that of the random variables

$$h_{1,a}^{(k)} := \boldsymbol{a}^{(k)} \cdot \boldsymbol{x}_{1,a}, \qquad h_{2,a} := \boldsymbol{a}^{(1)} \cdot \boldsymbol{x}_{2,a}, \qquad \tilde{h}_{1,a}^{(k)} := \tilde{\boldsymbol{a}}^{(k)} \cdot \boldsymbol{x}_{1,a}, \qquad \tilde{h}_{2,a} := \tilde{\boldsymbol{a}}^{(1)} \cdot \boldsymbol{x}_{2,a}, \tag{29}$$

$$H^{(k)} := \boldsymbol{a}_{\mathcal{I}(k)}^{(k)} \cdot \boldsymbol{x}_\star^{(k)} + \boldsymbol{a}_{\mathcal{I}(0)}^{(k)} \cdot \boldsymbol{x}_\star^{(0)} + \xi^{(k)}, \qquad \tilde{H}^{(k)} := \tilde{\boldsymbol{a}}_{\mathcal{I}(k)}^{(k)} \cdot \boldsymbol{x}_\star^{(k)} + \tilde{\boldsymbol{a}}_{\mathcal{I}(0)}^{(k)} \cdot \boldsymbol{x}_\star^{(0)} + \tilde{\xi}^{(k)}, \tag{30}$$

with $\{\boldsymbol{a}^{(k)}, \tilde{\boldsymbol{a}}^{(k)}\}_{k=1}^{K}$ being a set of $N$-dimensional centered Gaussian vectors with i.i.d. elements of variance $1/N$, and $\xi^{(k)}, \tilde{\xi}^{(k)}$ being independent centered Gaussian random variables with variance $(\sigma^{(k)})^2$ for $k = 1, \cdots, K$. Note that the random variables with different index $k$ are independent of each other. Following this observation, $\mathcal{L}$ can be expressed alternatively as $\mathcal{L} := \prod_{k=1}^{K} (\mathbb{E} L^{(k)})^{M^{(k)}} (\mathbb{E} \tilde{L}^{(k)})^{M^{(k)}}$, where

$$L^{(k)} = \prod_{a=1}^{n} \exp \left[ -\frac{\beta_1}{2}(H^{(k)} - h_{1,a}^{(k)})^2 - \frac{\delta_{1k}\beta_2}{2}(H^{(1)} - \kappa h_{1,a}^{(1)} - h_{2,a})^2 \right], \tag{31}$$

$$\tilde{L}^{(k)} = \prod_{a=1}^{n} \exp \left[ -t_1(\tilde{H}^{(k)} - \tilde{h}_{1,a}^{(k)})^2 - \delta_{1k} t_2(\tilde{H}^{(1)} - \kappa \tilde{h}_{1,a}^{(1)} - \tilde{h}_{2,a})^2 \right]. \tag{32}$$

Here, $\delta_{ij} = \mathbb{I}[i = j]$ denotes the Kronecker delta, which is unity if $i = j$ and zero otherwise. From the Gaussian nature of $\{\boldsymbol{a}^{(k)}, \tilde{\boldsymbol{a}}^{(k)}\}_{k=1}^{K}$ and $\{\xi^{(k)}, \tilde{\xi}^{(k)}\}_{k=1}^{K}$, the random variables $\{h_{1,a}^{(k)}\}_{a,k=1}^{n,K}$, $\{h_{2,a}\}_{a=1}^{n}$, and $\{H^{(k)}\}_{k=1}^{K}$ are

all centered Gaussians with the following covariance structure:

$$
\mathbb{E}[h_{1,a}^{(k)}h_{1,b}^{(l)}] = \delta_{kl}Q_1^{ab}, \qquad Q_1^{ab} := \frac{1}{N}\boldsymbol{x}_{1,a}\cdot\boldsymbol{x}_{1,b} = \frac{1}{N}\left(\sum_{k'=0}^{K}\boldsymbol{x}_{1,a}^{(k')}\cdot\boldsymbol{x}_{1,b}^{(k')} + \boldsymbol{x}_{1,a}^{(\text{neg.})}\cdot\boldsymbol{x}_{1,b}^{(\text{neg.})}\right),
$$

$$
\mathbb{E}[h_{2,a}h_{2,b}] = Q_2^{ab}, \qquad Q_2^{ab} := \frac{1}{N}\boldsymbol{x}_{2,a}\cdot\boldsymbol{x}_{2,b} = \frac{1}{N}\left(\sum_{k'=0}^{K}\boldsymbol{x}_{2,a}^{(k')}\cdot\boldsymbol{x}_{2,b}^{(k')} + \boldsymbol{x}_{2,a}^{(\text{neg.})}\cdot\boldsymbol{x}_{2,b}^{(\text{neg.})}\right),
$$

$$
\mathbb{E}[h_{1,a}^{(k)}h_{2,a}] = \delta_{1k}Q_r^{ab}, \qquad Q_r^{ab} := \frac{1}{N}\boldsymbol{x}_{1,a}\cdot\boldsymbol{x}_{2,b} = \frac{1}{N}\left(\sum_{k'=0}^{K}\boldsymbol{x}_{1,a}^{(k')}\cdot\boldsymbol{x}_{2,b}^{(k')} + \boldsymbol{x}_{1,a}^{(\text{neg.})}\cdot\boldsymbol{x}_{2,b}^{(\text{neg.})}\right), \tag{33}
$$

$$
\mathbb{E}[h_{1,a}^{(k)}H^{(l)}] = \delta_{kl}\big(m_1^{(k)}+m_1^{(0)}\big), \qquad m_{1,a}^{(k')} := \frac{1}{N}\boldsymbol{x}_{1,a}^{(k')}\cdot\boldsymbol{x}_\star^{(k')} \quad (k'=0,1,\cdots,K),
$$

$$
\mathbb{E}[h_{2,a}H^{(k)}] = \delta_{k1}\big(m_2^{(k)}+m_2^{(0)}\big), \qquad m_{2,a}^{(k')} := \frac{1}{N}\boldsymbol{x}_{2,a}^{(k')}\cdot\boldsymbol{x}_\star^{(k')} \quad (k'=0,1,\cdots,K),
$$

$$
\mathbb{E}[H^{(k)}H^{(l)}] = \delta_{kl}\left[\frac{1}{N}\|\boldsymbol{x}_\star^{(k)}\|_2^2 + \frac{1}{N}\|\boldsymbol{x}_\star^{(0)}\|_2^2 + (\sigma^{(k)})^2\right] = \delta_{kl}\rho^{(k)},
$$

for $1 \le k,l \le K$ and $1 \le a,b \le n$. Here, $\boldsymbol{x}_{1,a}^{(k)}$ and $\boldsymbol{x}_{2,a}^{(k)}$ are defined as the subvectors of $\boldsymbol{x}_{1,a}$ and $\boldsymbol{x}_{2,a}$ indexed by $\mathcal{I}(k)$ respectively, while $\boldsymbol{x}_{1,a}^{(\text{neg.})}$ and $\boldsymbol{x}_{2,a}^{(\text{neg.})}$ are both subvectors of $\boldsymbol{x}_{1,a}$ and $\boldsymbol{x}_{2,a}$ whose indices are not included in $\bigcup_{k=0}^{K}\mathcal{I}(k)$. The same covariance structure also follows for the random variables $\{\tilde{h}_{1,a}^{(k)}\}_{a,k=1}^{n,K}, \{\tilde{h}_{2,a}\}_{a=1}^{n}$, and $\{\tilde{H}^{(k)}\}_{k=1}^{K}$. Define the average with respect to the random variables given in equation 33, conditioned on $\Omega = \{Q_1^{ab}, Q_2^{ab}, Q_r^{ab}, m_{1,a}^{(k)}, m_{2,a}^{(k)}\}$, as $\mathbb{E}_{|\Omega}$. Inserting the trivial identities based on the delta function corresponding to the definitions of $\Omega$ given in equation 33, we can rewrite equation 28 up to a trivial multiplicative constant as

$$
\int d\Omega \int \left(\prod_{a=1}^{n}\prod_{k=0}^{K}d\boldsymbol{x}_{1,a}^{(k)}d\boldsymbol{x}_{2,a}^{(k)}\right)\exp\left[-\sum_{a=1}^{n}\Big(\beta_1\lambda_1\|\boldsymbol{x}_{1,a}\|_1 + \beta_2 r(\boldsymbol{x}_{1,a}|\boldsymbol{x}_{2,a})\Big) + \sum_{k=1}^{K}M^{(k)}\log(\mathbb{E}_{|\Omega}L^{(k)})(\mathbb{E}_{|\Omega}\tilde{L}^{(k)})\right]
$$

$$
\times\prod_{a\le b=1}^{n}\delta\left(NQ_1^{ab} - \sum_{k'=0}^{K}\boldsymbol{x}_{1,a}^{(k')}\cdot\boldsymbol{x}_{1,b}^{(k')} - \boldsymbol{x}_{1,a}^{(\text{neg.})}\cdot\boldsymbol{x}_{1,b}^{(\text{neg.})}\right)\delta\left(NQ_2^{ab} - \sum_{k'=0}^{K}\boldsymbol{x}_{2,a}^{(k')}\cdot\boldsymbol{x}_{2,b}^{(k')} - \boldsymbol{x}_{2,a}^{(\text{neg.})}\cdot\boldsymbol{x}_{2,b}^{(\text{neg.})}\right)
$$

$$
\times\prod_{a,b=1}^{n}\delta\left(NQ_r^{ab} - \sum_{k'=0}^{K}\boldsymbol{x}_{1,a}^{(k')}\cdot\boldsymbol{x}_{2,b}^{(k')} - \boldsymbol{x}_{1,a}^{(\text{neg.})}\cdot\boldsymbol{x}_{2,b}^{(\text{neg.})}\right)\mathbb{E}_\star\prod_{a=1}^{n}\prod_{k=0}^{K}\delta\big(Nm_{1,a}^{(k)} - \boldsymbol{x}_{1,a}^{(k)}\cdot\boldsymbol{x}_\star^{(k)}\big)\delta\big(Nm_{2,a}^{(k)} - \boldsymbol{x}_{2,a}^{(k)}\cdot\boldsymbol{x}_\star^{(k)}\big),
$$
$$\tag{34}$$

where $\mathbb{E}_\star$ is the average with respect to the set of ground truth vectors $\{\boldsymbol{x}_\star^{(k)}\}_{k=0}^{K}$, whose elements are i.i.d. according to a standard normal distribution. Note that the delta functions can be expressed alternatively using its Fourier representation as

$$
\delta\left(NQ_1^{ab} - \sum_{k'=0}^{K}\boldsymbol{x}_{1,a}^{(k')}\cdot\boldsymbol{x}_{1,b}^{(k')} - \boldsymbol{x}_{1,a}^{(\text{neg.})}\cdot\boldsymbol{x}_{1,b}^{(\text{neg.})}\right) = \int_{\mathbb{C}}d\hat{Q}_1^{ab}e^{\hat{Q}_1^{ab}\left(NQ_1^{ab}-\sum_{k'=0}^{K}\boldsymbol{x}_{1,a}^{(k')}\cdot\boldsymbol{x}_{1,b}^{(k')}-\boldsymbol{x}_{1,a}^{(\text{neg.})}\cdot\boldsymbol{x}_{1,b}^{(\text{neg.})}\right)},
$$

$$
\delta\left(NQ_2^{ab} - \sum_{k'=0}^{K}\boldsymbol{x}_{2,a}^{(k')}\cdot\boldsymbol{x}_{2,b}^{(k')} - \boldsymbol{x}_{2,a}^{(\text{neg.})}\cdot\boldsymbol{x}_{2,b}^{(\text{neg.})}\right) = \int_{\mathbb{C}}d\hat{Q}_2^{ab}e^{\hat{Q}_2^{ab}\left(NQ_2^{ab}-\sum_{k'=0}^{K}\boldsymbol{x}_{2,a}^{(k')}\cdot\boldsymbol{x}_{2,b}^{(k')}-\boldsymbol{x}_{2,a}^{(\text{neg.})}\cdot\boldsymbol{x}_{2,b}^{(\text{neg.})}\right)},
$$

$$
\delta\left(NQ_r^{ab} - \sum_{k'=0}^{K}\boldsymbol{x}_{1,a}^{(k')}\cdot\boldsymbol{x}_{2,b}^{(k')} - \boldsymbol{x}_{1,a}^{(\text{neg.})}\cdot\boldsymbol{x}_{2,b}^{(\text{neg.})}\right) = \int_{\mathbb{C}}d\hat{Q}_r^{ab}e^{\hat{Q}_r^{ab}\left(NQ_r^{ab}-\sum_{k'=0}^{K}\boldsymbol{x}_{1,a}^{(k')}\cdot\boldsymbol{x}_{2,b}^{(k')}-\boldsymbol{x}_{1,a}^{(\text{neg.})}\cdot\boldsymbol{x}_{2,b}^{(\text{neg.})}\right)}, \tag{35}
$$

$$
\delta\big(Nm_{1,a}^{(k)} - \boldsymbol{x}_{1,a}^{(k)}\cdot\boldsymbol{x}_\star^{(k)}\big) = \int_{\mathbb{C}}d\hat{m}_{1,a}^{(k)}e^{\hat{m}_{1,a}^{(k)}\left(Nm_{1,a}^{(k)}-\boldsymbol{x}_{1,a}^{(k)}\cdot\boldsymbol{x}_\star^{(k)}\right)},
$$

$$
\delta\big(Nm_{2,a}^{(k)} - \boldsymbol{x}_{2,a}^{(k)}\cdot\boldsymbol{x}_\star^{(k)}\big) = \int_{\mathbb{C}}d\hat{m}_{2,a}^{(k)}e^{\hat{m}_{2,a}^{(k)}\left(Nm_{2,a}^{(k)}-\boldsymbol{x}_{2,a}^{(k)}\cdot\boldsymbol{x}_\star^{(k)}\right)},
$$

up to a trivial multiplicative constant.

To obtain an expression that is analytically continuable to $n \to 0$, we introduce the replica symmetric ansatz (Charbonneau et al., 2023), which assumes that the integral over $\Omega$ is dominated by the contribution of the subspace where $\Omega$ satisfy the following constraints:

$$Q_1^{ab} = q_1 + (1 - \delta_{ab})\frac{\chi_1}{\beta_1}, \qquad Q_2^{ab} = q_2 + (1 - \delta_{ab})\frac{\chi_2}{\beta_2}, \qquad Q_r^{ab} = q_r + (1 - \delta_{ab})\frac{\chi_r}{\beta_1},$$
$$m_{1,a}^{(k)} = m_1^{(k)}, \qquad m_{2,a}^{(k)} = m_2^{(k)}, \tag{36}$$

for $1 \le a, b \le n$ and $0 \le k \le K$. Although a general proof of this ansatz itself is still lacking, calculations based on this assumption are known to be asymptotically exact in the limit $N \to \infty$ for convex generalized linear models (Gerbelot et al., 2023) and Bayes-optimal estimation (Barbier et al., 2019). Consequently, the conjugate variables $\{\hat{Q}_1^{ab}, \hat{Q}_2^{ab}, \hat{Q}_r^{ab}, \hat{m}_{1,a}^{(k)}, \hat{m}_{2,a}^{(k)}\}$ are also assumed to be replica symmetric:

$$\hat{Q}_1^{ab} = \beta_1 \hat{q}_1 - (1 - \delta_{ab})\beta_1^2 \hat{\chi}_1, \qquad \hat{Q}_2^{ab} = \beta_2 \hat{q}_2 - (1 - \delta_{ab})\beta_2^2 \hat{\chi}_2, \qquad \hat{Q}_r^{ab} = -\beta_2 \hat{q}_r + (1 - \delta_{ab})\beta_1 \beta_2 \hat{\chi}_r,$$
$$\hat{m}_{1,a}^{(k)} = -\beta_1 \hat{m}_1^{(k)}, \qquad \hat{m}_{2,a}^{(k)} = -\beta_2 \hat{m}_2^{(k)}. \tag{37}$$

Inserting equations 36 and 37 to equation 34, and rewriting $\mathbb{E}_{|\Omega}$ as $\mathbb{E}_{|\Theta}$ given this simplified profile of $\Omega$ offers

$$\mathbb{E}_\star \int_\mathbb{C} d\Theta_1 d\Theta_2 \int \left( \prod_{a=1}^n \prod_{k=0}^K dx_{1,a}^{(k)} dx_{2,a}^{(k)} \right) \exp \left[ \sum_{k=1}^K M^{(k)} \log(\mathbb{E}_{|\Theta} L^{(k)}) + \sum_{k=1}^K M^{(k)} \log(\mathbb{E}_{|\Theta} \tilde{L}^{(k)}) \right]$$

$$\times \exp nN \left[ \beta_1 \left( \frac{q_1 \hat{q}_1 - \chi_1 \hat{\chi}_1}{2} - \sum_{k=0}^K m_1^{(k)} \hat{m}_1^{(k)} \right) + \beta_2 \left( \frac{q_2 \hat{q}_2 - \chi_2 \hat{\chi}_2}{2} - q_r \hat{q}_r - \chi_r \hat{\chi}_r - \sum_{k=0}^K m_2^{(k)} \hat{m}_2^{(k)} \right) \right]$$

$$\times \exp \left[ -\beta_1 \sum_{a=1}^n \sum_{k=0}^K \left( \frac{\hat{q}_1}{2} \|\boldsymbol{x}_{1,a}^{(k)}\|_2^2 - \hat{m}_1 \boldsymbol{x}_\star^{(k)} \cdot \boldsymbol{x}_{1,a}^{(k)} + \lambda_1 \|\boldsymbol{x}_{1,a}^{(k)}\|_1 \right) - \beta_1 \sum_{a=1}^n \left( \frac{\hat{q}_1}{2} \|\boldsymbol{x}_{1,a}^{(\text{neg.})}\|_2^2 + \lambda_1 \|\boldsymbol{x}_{1,a}^{(\text{neg.})}\|_1 \right) \right.$$

$$- \beta_2 \sum_{a=1}^n \sum_{k=0}^K \left( \frac{\hat{q}_2}{2} \|\boldsymbol{x}_{2,a}^{(k)}\|_2^2 - (\hat{m}_2 \boldsymbol{x}_\star^{(k)} + \hat{q}_r \boldsymbol{x}_{1,a}^{(k)}) \cdot \boldsymbol{x}_{2,a}^{(k)} + r(\boldsymbol{x}_{2,a}^{(k)} | \boldsymbol{x}_{1,a}^{(k)}) \right)$$

$$- \beta_2 \sum_{a=1}^n \left( \frac{\hat{q}_2}{2} \|\boldsymbol{x}_{2,a}^{(\text{neg.})}\|_2^2 - \hat{q}_r \boldsymbol{x}_{1,a}^{(\text{neg.})} \cdot \boldsymbol{x}_{2,a}^{(\text{neg.})} + r(\boldsymbol{x}_{2,a}^{(\text{neg.})} | \boldsymbol{x}_{1,a}^{(\text{neg.})}) \right)$$

$$\left. + \frac{\beta_1^2 \hat{\chi}_1}{2} \left\| \sum_{a=1}^n \boldsymbol{x}_{1,a} \right\|_2^2 + \frac{\beta_2^2 \hat{\chi}_2}{2} \left\| \sum_{a=1}^n \boldsymbol{x}_{2,a} \right\|_2^2 + \beta_1 \beta_2 \hat{\chi}_r \left( \sum_{a=1}^n \boldsymbol{x}_{1,a} \right) \cdot \left( \sum_{a=1}^n \boldsymbol{x}_{2,a} \right) \right]. \tag{38}$$

The last equation can be further simplified using the equality

$$\exp \left[ \frac{\beta_1^2 \hat{\chi}_1}{2} \left\| \sum_{a=1}^n \boldsymbol{x}_{1,a} \right\|_2^2 + \frac{\beta_2^2 \hat{\chi}_2}{2} \left\| \sum_{a=1}^n \boldsymbol{x}_{2,a} \right\|_2^2 + \beta_1 \beta_2 \hat{\chi}_r \left( \sum_{a=1}^n \boldsymbol{x}_{1,a} \right) \cdot \left( \sum_{a=1}^n \boldsymbol{x}_{2,a} \right) \right] \tag{39}$$

$$= \mathbb{E}_{\boldsymbol{z}_1, \boldsymbol{z}_2} \exp \left[ \beta_1 \sqrt{\hat{\chi}_1} \sum_{a=1}^n \boldsymbol{z}_1 \cdot \boldsymbol{x}_{1,a} + \beta_2 \sqrt{\hat{\chi}_2} \sum_{a=1}^n \boldsymbol{z}_2 \cdot \boldsymbol{x}_{2,a} \right] \tag{40}$$

where $\boldsymbol{z}_1, \boldsymbol{z}_2 \in \mathbb{R}^N$ are random Gaussian vectors with elements i.i.d. according to $z_{1,i} \sim \mathsf{z}_1, z_{2,i} \sim \mathsf{z}_2$ for all $i = 1, \cdots, N$. From this decomposition, the integrals over $\boldsymbol{x}_{1,a}$ and $\boldsymbol{x}_{2,a}$ decouple over both vector coordinates $i = 1, \cdots, N$ and replica indices $a = 1, \cdots, n$:

$$\int d\Theta_1 d\Theta_2 \int e^{\sum_{k=1}^K M^{(k)} \log \mathbb{E}_{|\Theta} L^{(k)} + \sum_{k=1}^K M^{(k)} \log \mathbb{E}_{|\Theta} \tilde{L}^{(k)}} \prod_{k=0}^K \left\{ \mathbb{E}_2 \left[ \int dx_1 dx_2 e^{-\beta_1 \mathsf{E}_1^{(k)}(x_1) - \beta_2 \mathsf{E}_2^{(k)}(x_2 | x_1)} \right]^n \right\}^{N^{(k)}}$$

$$\times \left\{ \mathbb{E}_2 \left[ \int dx_1 dx_2 e^{-\beta_1 \mathsf{E}_1^{(\text{neg.})}(x_1) - \beta_2 \mathsf{E}_2^{(\text{neg.})}(x_2 | x_1)} \right]^n \right\}^{N - \sum_{k=0}^K N^{(k)}}. \tag{41}$$

In the successive limit of $\beta_1 \to \infty$ and $\beta_2 \to \infty$, the integrals over $x_1$ and $x_2$ can be evaluated using Laplace's method, yielding the asymptotic form:

$$
\int d\Theta_1 d\Theta_2 \prod_{k=1}^{K} e^{M^{(k)} \log \mathbb{E}_{|\Theta} L^{(k)} + M^{(k)} \log \mathbb{E}_{|\Theta} \tilde{L}^{(k)}} \prod_{k=0}^{K} \left\{ \mathbb{E}_2 \left[ e^{-\beta_1 \mathsf{E}_1^{(k)}(\mathsf{x}_1^{(k)}) - \beta_2 \mathsf{E}_2^{(k)}(\mathsf{x}_2^{(k)} | \mathsf{x}_1^{(k)})} \right]^n \right\}^{N^{(k)}}
$$
$$
\times \left\{ \mathbb{E}_2 \left[ e^{-\beta_1 \mathsf{E}_1^{(\mathrm{neg.})}(\mathsf{x}_1^{(\mathrm{neg.})}) - \beta_2 \mathsf{E}_2^{(\mathrm{neg.})}(\mathsf{x}_2^{(\mathrm{neg.})} | \mathsf{x}_1^{(\mathrm{neg.})})} \right]^n \right\}^{N - \sum_{k=0}^{K} N^{(k)}}.
\tag{42}
$$

where the definitions of $\mathbb{E}_2$ and $\{\mathsf{x}_1^{(k)}, \mathsf{x}_2^{(k)}\}_{k=0}^{K}, \mathsf{x}_1^{(\mathrm{neg.})}, \mathsf{x}_2^{(\mathrm{neg.})}$ follow from Definitions 1 and 2.

Let us proceed with the calculation of $\mathbb{E}_{|\Theta} L^{(k)}$. Define the centered Gaussian variables $\{H^{(k)}, h_1^{(k)}, h_2^{(k)}\}$ and $\{u_{1,a}, u_{2,a}\}_{a=1}^{n}$ as

$$
\begin{pmatrix} H^{(k)} \\ h_1^{(k)} \\ h_2^{(k)} \end{pmatrix} \sim \mathcal{N}\left( \mathbf{0}_3, \begin{pmatrix} \rho^{(k)} & m_1^{(k)} & m_2^{(k)} \\ m_1^{(k)} & q_1 & q_r \\ m_2^{(k)} & q_r & q_2 \end{pmatrix} \right), \qquad \begin{pmatrix} u_{1,a} \\ u_{2,a} \end{pmatrix} \sim \mathcal{N}\left( \mathbf{0}_2, \begin{pmatrix} \chi_1/\beta_1 & \chi_r/\beta_1 \\ \chi_r/\beta_1 & \chi_2/\beta_2 \end{pmatrix} \right),
\tag{43}
$$

where $\{u_{1,a}, u_{2,a}\}$ are independent for all $a = 1, \cdots, n$. Then, we can see that the random variables admit the decomposition $h_{1,a}^{(k)} = h_1^{(k)} + u_{1,a}, h_{2,a} = h_2^{(k)} + u_{2,a}$, offering the expression

$$
\mathbb{E}_{|\Theta} L^{(k)} = \mathbb{E}_{h_1^{(k)}, h_2^{(k)}, H^{(k)}} \prod_{a=1}^{n} \mathbb{E}_{u_{1,a}, u_{2,a}} \exp\left[ -\frac{\beta_1}{2} (H^{(k)} - h_1^{(k)} - u_{1,a})^2 - \frac{\beta_2 \delta_{1k}}{2} \left( H^{(k)} - \kappa(h_1^{(k)} + u_{1,a}) - h_2^{(k)} - u_{2,a} \right)^2 \right].
\tag{44}
$$

Taking into account that $\beta_1 \gg \beta_2$, the Gaussian measure over $(u_{1,a}, u_{2,a})$ can be written as

$$
C_{\beta_1, \beta_2} \exp\left[ -\frac{\beta_1}{2\chi_1} (1 + c_{\beta_1}) u_{1,a}^2 - \frac{\beta_2}{2\chi_2} \left( u_{2,a} - \frac{\chi_r}{\chi_1} u_{1,a} \right)^2 \right] du_{1,a} du_{2,a},
\tag{45}
$$

where $C_{\beta_1, \beta_2}$ is a number subexponential in $\beta_1$ and $\beta_2$, and $c_{\beta_1}$ is a real number which converges to zero as $\beta_1 \to \infty$. Using this expression to equation 44 yields

$$
\mathbb{E}_{|\Theta} L^{(k)} = \mathbb{E}_{h_1^{(k)}, h_2^{(k)}, H^{(k)}} \left\{ \int du_1 du_2 \exp\left[ -\frac{\beta_1}{2\chi_1} (1 + c_{\beta_1}) u_1^2 - \frac{\beta_1}{2} (H^{(k)} - h_1^{(k)} - u_1)^2 \right.\right.
$$
$$
\left.\left. -\frac{\beta_2}{2\chi_2} u_2^2 - \frac{\beta_2 \delta_{1k}}{2} \left( H^{(k)} - \kappa(h_1^{(k)} + u_1) - h_2^{(k)} - u_2 - \frac{\chi_r}{\chi_1} u_1 \right)^2 + \log C_{\beta_1, \beta_2} \right] \right\}^n.
\tag{46}
$$

The same procedure can be applied to $\tilde{L}^{(k)}$, this time accounting for $t_1, t_2 \ll \beta_2$, offering

$$
\mathbb{E}_{|\Theta} \tilde{L}^{(k)} = \mathbb{E}_{\tilde{h}_1^{(k)}, \tilde{h}_2^{(k)}, \tilde{H}^{(k)}} \exp n \left[ -t_1 \left( \tilde{H}^{(k)} - \tilde{h}_1^{(k)} \right)^2 - t_2 \delta_{1k} \left( \tilde{H}^{(k)} - \kappa \tilde{h}_1^{(k)} - \tilde{h}_2^{(k)} \right)^2 + \log C_{\beta_1, \beta_2} \right].
\tag{47}
$$

Note that the expressions for $\tilde{L}^{(k)}$ and $L^{(k)}$ are simple Gaussian integrals that can be computed explicitly. Now that all the terms in equation 34 have been expressed in analytical form with respect to $n$, the limit $n \to 0$ can be taken formally. Evaluating equations 42, 46 and 47 up to first order of $n$, and neglecting subleading terms with respect to $\beta_1$ and $\beta_2$, we finally obtain an expression of the form

$$
\mathbb{E}_{\mathcal{D}, \tilde{\mathcal{D}}} \left[ \Phi^n(\mathcal{D}, \tilde{\mathcal{D}}) \right] = \int d\Theta_1 d\Theta_2 \exp N \left[ n\mathcal{G}(\Theta_1, \Theta_2) + O(n^2) \right],
\tag{48}
$$

where

$$
\mathcal{G}(\Theta_1, \Theta_2) = \beta_1 G_1(\Theta_1) + \beta_2 G_2(\Theta_2 | \Theta_1) + t_1 \epsilon_1 + t_2 \epsilon_2,
\tag{49}
$$

$$G_1(\Theta_1) = \frac{q_1 \hat{q}_1 - \chi_1 \hat{\chi}_1}{2} - \sum_{k=0}^{K} m_1^{(k)} \hat{m}_1^{(k)} - \sum_{k=0}^{K} \pi^{(k)} \mathbb{E}_1 \left[ \mathsf{E}_1^{(k)}\left(\mathsf{x}_1^{(k)}\right) \right] - \pi^{(\text{neg.})} \mathbb{E}_1 \left[ \mathsf{E}_1^{(\text{neg.})}\left(\mathsf{x}_1^{(\text{neg.})}\right) \right]$$

$$- \sum_{k=1}^{K} \frac{\alpha^{(k)}}{2} \frac{q_1 - 2(m_1^{(0)} + m_1^{(k)}) + \rho^{(k)}}{1 + \chi_1}, \tag{50}$$

$$G_2(\Theta_2|\Theta_1) = \frac{q_2 \hat{q}_2 - \chi_2 \hat{\chi}_2}{2} - q_r \hat{q}_r - \chi_r \hat{\chi}_r - \sum_{k=0}^{K} m_2^{(k)} \hat{m}_2^{(k)}$$

$$- \sum_{k=0}^{K} \pi^{(k)} \mathbb{E}_2 \left[ \mathsf{E}_2^{(k)}\left(\mathsf{x}_2^{(k)}|\mathsf{x}_1^{(k)}\right) \right] - \pi^{(\text{neg.})} \mathbb{E}_2 \left[ \mathsf{E}_2^{(\text{neg.})}\left(\mathsf{x}_2^{(\text{neg.})}|\mathsf{x}_1^{(\text{neg.})}\right) \right]$$

$$- \frac{\alpha^{(1)}}{2} \frac{q_2 + A^2 q_1 + 2A q_r + B^2 \rho^{(1)} - 2B(m_2^{(0)} + m_2^{(1)}) - 2AB(m_1^{(0)} + m_1^{(1)})}{1 + \chi_2}, \tag{51}$$

and

$$\epsilon_1 = \sum_{k=1}^{K} \pi^{(k)} \left[ q_1 - 2(m_1^{(0)} + m_1^{(k)}) + \rho^{(k)} \right], \tag{52}$$

$$\epsilon_2 = \pi^{(1)} \left[ q_2 + \kappa^2 q_1 + 2\kappa q_r + \rho^{(1)} - 2(m_2^{(0)} + m_2^{(1)}) - 2\kappa(m_1^{(0)} + m_1^{(1)}) \right]. \tag{53}$$

For large $N$ and finite $n$, the integral over $\Theta_1$ and $\Theta_2$ can be evaluated using the saddle point method, where the integral is dominated by the stationary point of the exponent. This finally yields

$$\lim_{n \to 0} \lim_{N \to \infty} \frac{1}{N} \frac{\partial}{\partial n} \log \mathbb{E}_{\mathcal{D}, \tilde{\mathcal{D}}} \left[ \Phi^n(\mathcal{D}, \tilde{\mathcal{D}}) \right] = \underset{\Theta_1, \Theta_2}{\mathrm{Extr}} \, \mathcal{G}(\Theta_1, \Theta_2), \tag{54}$$

where Extr denotes the extremum operation of the function. The stationary conditions for $\Theta_1$ and $\Theta_2$, in the successive limit of $\beta_1 \to \infty$ and $\beta_2 \to \infty$ are given by the equations of state in Definitions 1 and 2. Note that the term $t_1 \epsilon_1 + t_2 \epsilon_2$ does not contribute to the stationary condition in the limit $t_1, t_2 \to 0$, and thus can be neglected until one takes the derivative with respect to $t_1$ and $t_2$, as in equations 11 and 12.

## B   Additional numerical experiments

Here, we provide more numerical experiments highlighting the difference between the Generalized Trans-Lasso, Pretraining Lasso and Trans-Lasso. In figures 6 and 7, we show the ratios of $\min\{\epsilon_{\Delta\lambda=0}, \epsilon_{\kappa=0}\}$, $\epsilon_{\text{Pretrain}}$ and $\epsilon_{\text{Trans}}$ against $\epsilon_{\text{LO}}$ for the cases $(\pi^{(0)}, \pi^{(1)}, \pi^{(2)}) = (0.15, 0.04, 0.04)$ and $(0.05, 0.14, 0.14)$, respectively. The former case represents the problem setting where the underlying common feature vector among the datasets is large, while the latter represents the problem setting where the underlying common feature vector is small. In both cases, the minimum of the $\kappa = 0$ or $\Delta\lambda = 0$ strategy can obtain generalization errors close to the one from the LO strategy for both low or moderate noise levels. This is not a property exhibited in the Trans-Lasso or Pretraining Lasso, where both algorithms have relatively low generalization performance when $\sigma = 0.01$ and $(\pi^{(0)}, \pi^{(1)}, \pi^{(2)}) = (0.15, 0.04, 0.04)$. Note that when $(\pi^{(0)}, \pi^{(1)}, \pi^{(2)}) = (0.05, 0.14, 0.14)$, all three algorithms yield comparable generalization performance.

To directly compare the $\kappa = 0$ or $\Delta\lambda = 0$ strategy with the Pretraining Lasso and Trans-Lasso, in figures 8, 9 and 10, we also plot the same ratios with shared color scale bars. As evident from the plot, the $\kappa = 0$ or $\Delta\lambda = 0$ strategy exhibits a clear improvement in generalization performance over Trans-Lasso. While the difference between Pretraining Lasso and $\kappa = 0$ or $\Delta\lambda = 0$ strategy is comparable for $\sigma = 0.1$, we can see a consistent advantage for lower noise levels ($\sigma = 0.01$).

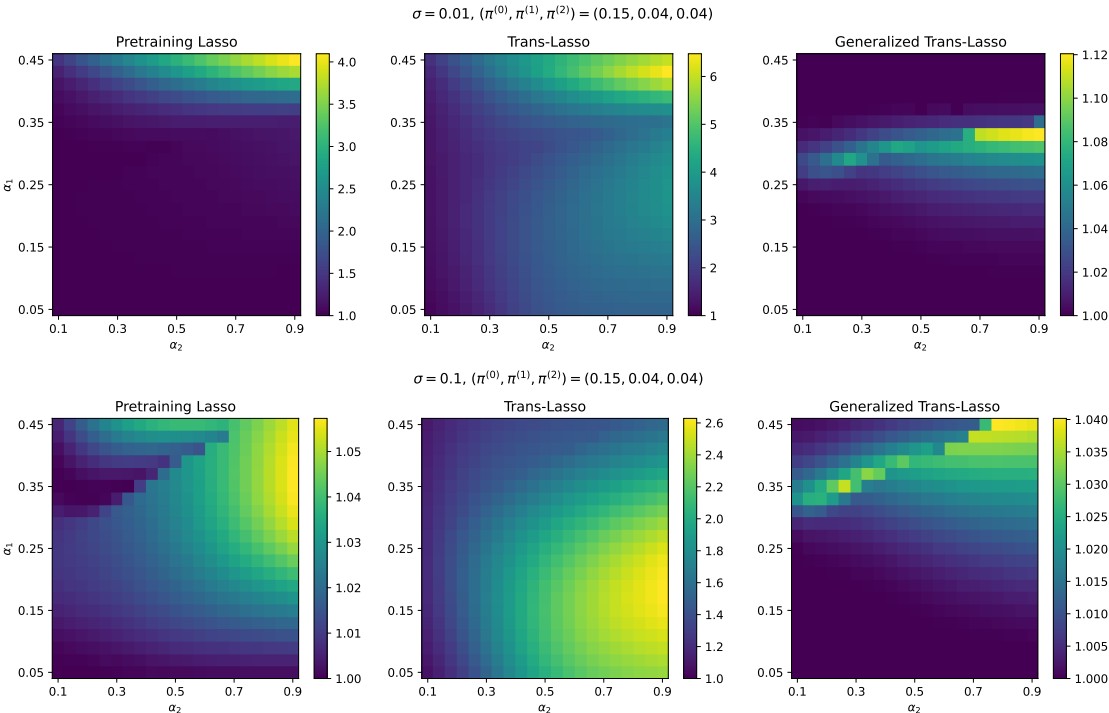

Figure 6: Ratios $\min\{\epsilon_{\Delta\lambda=0}, \epsilon_{\kappa=0}\}/\epsilon_{\mathrm{LO}}, \epsilon_{\mathrm{Pretrain}}/\epsilon_{\mathrm{LO}}$ and $\epsilon_{\mathrm{Trans}}/\epsilon_{\mathrm{LO}}$ under setting $(\pi^{(0)}, \pi^{(1)}, \pi^{(2)}) = (0.15, 0.04, 0.04)$ and noise level $\sigma = 0.01$ (top figure) and $0.1$ (bottom figure) for various values of $(\alpha^{(1)}, \alpha^{(2)})$.

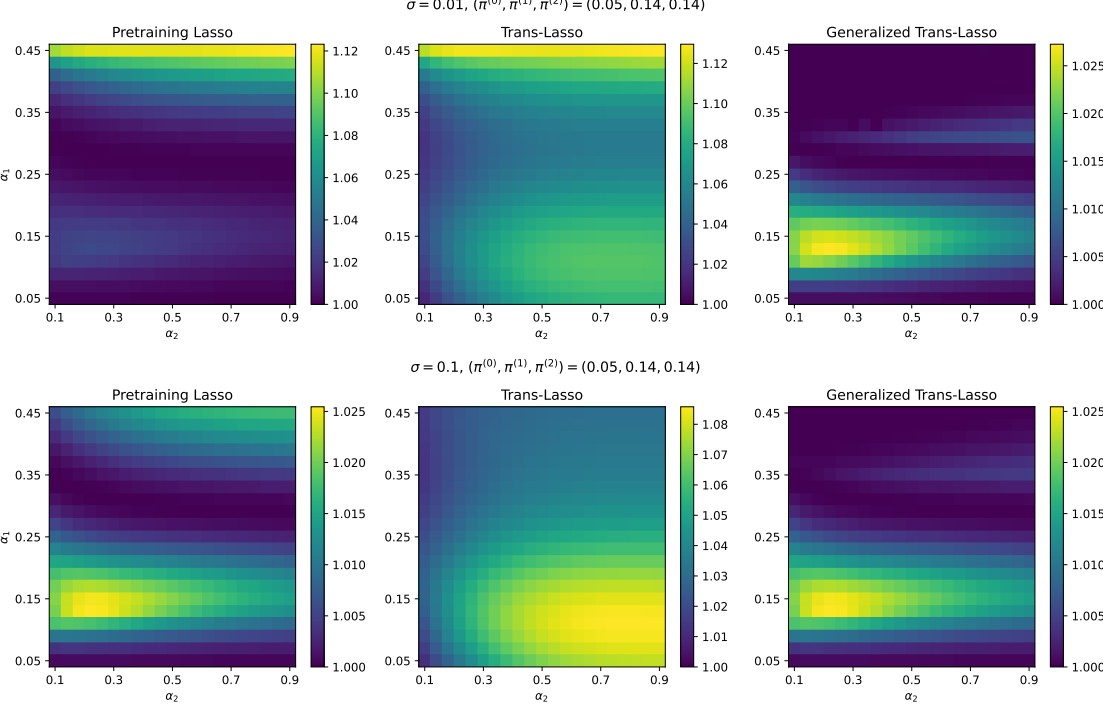

Figure 7: Ratios $\min\{\epsilon_{\Delta\lambda=0}, \epsilon_{\kappa=0}\}/\epsilon_{\mathrm{LO}}, \epsilon_{\mathrm{Pretrain}}/\epsilon_{\mathrm{LO}}$ and $\epsilon_{\mathrm{Trans}}/\epsilon_{\mathrm{LO}}$ under setting $(\pi^{(0)}, \pi^{(1)}, \pi^{(2)}) = (0.05, 0.14, 0.14)$ and noise level $\sigma = 0.01$ (top figure) and $0.1$ (bottom figure) for various values of $(\alpha^{(1)}, \alpha^{(2)})$.

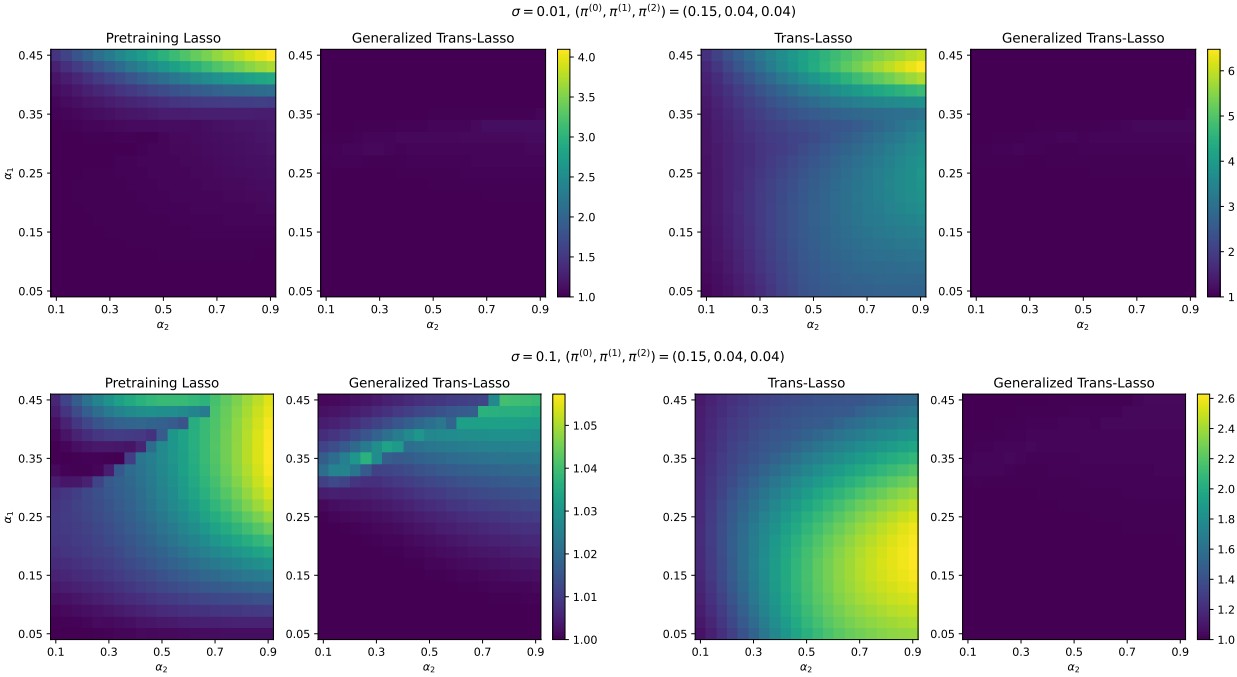

Figure 8: Comparisons between ratios $\min\{\epsilon_{\Delta\lambda=0}, \epsilon_{\kappa=0}\}/\epsilon_{\mathrm{LO}}$, $\epsilon_{\mathrm{Pretrain}}/\epsilon_{\mathrm{LO}}$ and $\epsilon_{\mathrm{Trans}}/\epsilon_{\mathrm{LO}}$ under setting $(\pi^{(0)}, \pi^{(1)}, \pi^{(2)}) = (0.15, 0.04, 0.04)$.

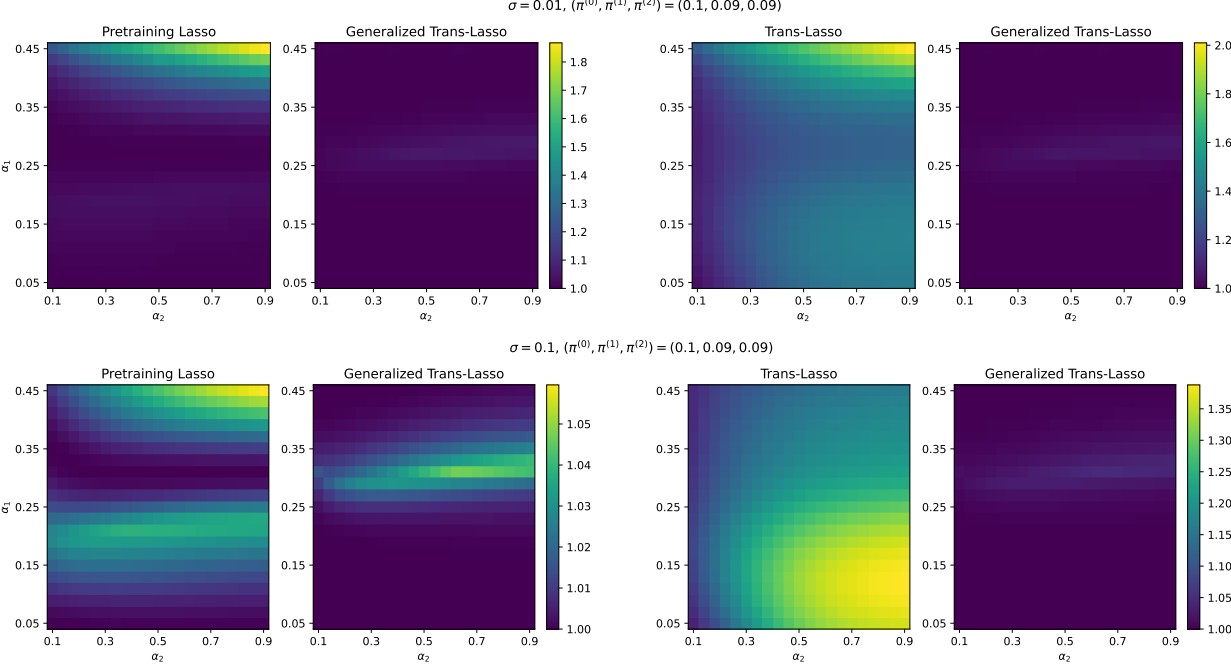

Figure 9: Comparisons between ratios $\min\{\epsilon_{\Delta\lambda=0}, \epsilon_{\kappa=0}\}/\epsilon_{\mathrm{LO}}$, $\epsilon_{\mathrm{Pretrain}}/\epsilon_{\mathrm{LO}}$ and $\epsilon_{\mathrm{Trans}}/\epsilon_{\mathrm{LO}}$ under setting $(\pi^{(0)}, \pi^{(1)}, \pi^{(2)}) = (0.10, 0.09, 0.09)$.

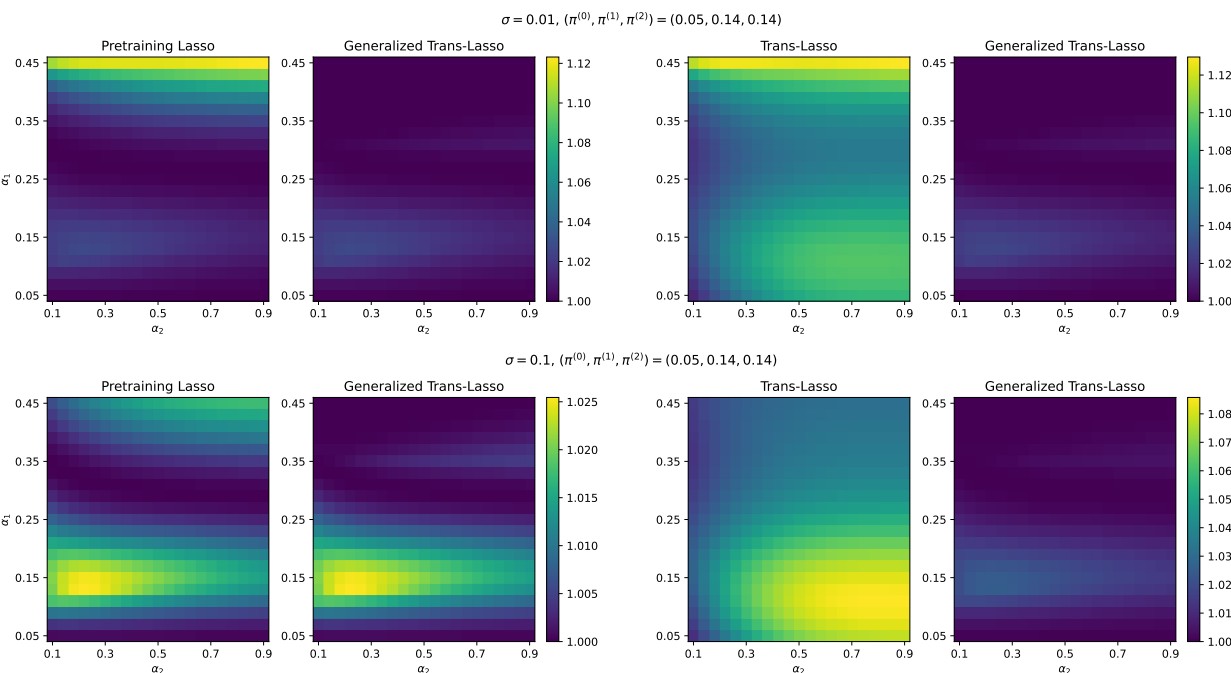

Figure 10: Comparisons between ratios $\min\{\epsilon_{\Delta\lambda=0}, \epsilon_{\kappa=0}\}/\epsilon_{\mathrm{LO}}$, $\epsilon_{\mathrm{Pretrain}}/\epsilon_{\mathrm{LO}}$ and $\epsilon_{\mathrm{Trans}}/\epsilon_{\mathrm{LO}}$ under setting $(\pi^{(0)}, \pi^{(1)}, \pi^{(2)}) = (0.05, 0.14, 0.14)$.

