# OpenReview forum: "Transfer Learning in $\ell_1$ Regularized Regression: Hyperparameter Selection Strategy based on Sharp Asymptotic Analysis"
_TMLR — Accepted by TMLR_

### Review · Reviewer_jR6M · 2024-10-29

**Summary Of Contributions:**

This manuscript focuses on the problem of transfer learning in sparse high-dimensional linear regression, and studies the hyperparameter selection aspect of some recent two-stage regression schemes. Proposing a generalization of two existing schemes, the paper pursues a direct relationship between the generalization error of this scheme (for both the stages) and the selected hyperparameters. For this purpose, the authors perform an asymptotic analysis under various assumptions (Gaussian design matrices and feature vectors) utilizing tools from statistical physics (the replica method), establishing a systems of equations whose solution lead to closed-form analytic expressions for the generalization error utilizing problem dependent quantities. Then the authors highlight how the estimated generalization error in a finite data setting closely matches the asymptotic estimate (given that the data satisfy the required assumptions and that various data dependent quantities are known), thereby motivating the asymptotic analysis.

Based on the insights from the finite data simulations, the author motivate a couple of simplified hyperparameter selection processes that remove the need to tune one of the 4 hyperparameters and empirically demonstrate the ability of these simplified strategies to match the more computationally expensive strategy of optimizing all 4 hyperparameters simultaneously. The empirical evaluations consider both synthetic data as well as the non-synthetic IMDB dataset. The empirical evaluations with the synthetic datasets highlight the different scenarios (data scarce vs data abundant or low noise vs high noise or low support overlap vs high support overlap) where the proposed simplified hyperparameter selection strategy perform competitively to the more expensive standard strategy. The experiments on the IMDB dataset highlight that the simplified strategies outperform the existing baselines.

**Audience:**

Yes

**Broader Impact Concerns:**

There is no broader impact discussion in this paper, but there are no ethical concerns with this work in my opinion.

**Claims And Evidence:**

Yes

**Requested Changes:**

Based on my discussion on the weaknesses, I think it is critical to address at least weakness W1.

**Strengths And Weaknesses:**

### Strengths:
- (S1) The paper focuses on the critical computational limitation of hyperparameter selection with a theoretically motivated heuristic to reduce the number of hyperparameters to be searched over.
- (S2) The empirical results in Section 4 demonstate how the generalization error estimated with finite data closely matches the asymptotic quantities, thereby nicely motivating the utility of an asymptotic analysis.
- (S3) The paper provides nice discussion of the empirical results, and thus, proposes well motivated heuristics for efficient hyperparameter selection strategies, considering different problem setups such as scarce vs abundant data or low vs high noise.

### Weaknesses:

Given my lack of expertise in statistical physics, and its application to machine learning, I believe that there are three main weaknesses of this manuscript:
- (W1) The main analysis leading to the expression for the generalization error lacks appropriate explanations, motivations and discussions (even in the appendix), making the presentation extremely convoluted, and leaving the reader with no intuition as to why the analysis is meaningful and of interest. The following are some specific instances of the above issue:
  - (W1.1) From the presentation of Section 3.1, there is no motivation for how we arrive at equations (11) and (12). Establishing a connection to (7) and (8) would be very useful. I can see the connection of $\beta_1 \to \infty$ and $\beta_2 \to \infty$ but the role of $t_1, t_2$ needs clear explanation
  - (W1.2) It is not clear from the description why we should care about the equation (13) and how it is related to (11) and (12). I can see that when applying the $\partial / \partial t_1$ and then applying $t_1 \to 0$, we get the term inside the integral but no $\log \left( \int \cdots \right)$. So I think it needs some clarification since the ensuing discussion following (13) highlights the role of the replica method in handling the right hand side of equation (13). Even the detailed version in Appendix A begins by trying to compute the right-hand-side of equation (13) except that the equation (23) actually contains the exponentiated parts of equations (11) and (12) while the right hand side of (13) did not. So that is already a discrepancy that needs to be clarified.
  - (W1.3) For the untrained eye, it might be useful to clarify why the equality in equation (13) is true.
  - (W1.4) The "equations of state" in definition 1 need a lot of clarification. There are many unclear and somewhat confusing elements to it. Without proper motivation or understanding, I dont think readers will be able to understand why such a result makes sense and how they might be extended to situations where the assumptions of the analysis do not hold, which makes the utility of such analysis quite limited.
    - (W1.4.1) First, the variables $\Theta_1$ need some introduction.
    - (W1.4.2) Second, we state that the $\mathsf{x}_1^{(k)}$ etc are obtained as a solution to a random optimization problems (14) and (15) given $\Theta_1$. However, then equation (16) defines the system of equations driving the variables $\Theta_1$, which depends on (the expectation of) the solutions to (14) and (15). This bilevel nature of this system of equation makes the definition quite confusing without appropriate motivation and discussion.
    - (W1.4.3) Given the bilevel structure of this system of equations (16), it is also good to understand the conditions under which we should expect a solution. Alternately, it would be good to understand the salient structure of this system of equations that ensures there will always be a solution. Similarly, it might be good to motivate why we are guaranteed to find finite solutions and why certain variables are ensured to be nonnegative (for example, why is $\hat{\chi}_1$ guaranteed to be positive in equations (14) and (15)).
    - (W1.4.4) Note that some of the definitions require the gradient of an optimal solution (for example, $\frac{\partial}{\partial \mathsf{z}_1} \arg \min_x \mathsf{E}_1^{(k)}(x)$) which is a nontrivial quantity in general.
  - (W1.5) Similar lack of clarity and motivation for definition 2
  - (W1.6) - Given the quantities defined in definitions 1 and 2, it is nice that the generalization errors are derived from simple expressions. However, without proper introduction and discussion of the quantities obtained via definitions 1 and 2, the results are not intuitive -- this would have been a great opportunity to highlight how the generalization error decomposes into different (possibly intuitive) terms, and how each of these terms are affected by the hyperparameters.
- (W2) The definitions of the generalization error being considered in this manuscript seem unintuitive, and thus closed-form expression for such definitions makes the connection to the more standard definitions of generalization unclear.
  - (W2.1) In equations (7) and (8), it is not clear why we are scaling by number of features $N$ to define the generalization errors. I think we should be scaling by $(1/K)$ (for just equation (7)) and the number of observations $\tilde{M}_k$ per new dataset $\tilde{\mathcal{D}}^{(k)}$ (for both (7) and (8)). So it could be something like the per-dataset mean-squared error. Equation (7) could be

$$\epsilon^{\text{(1st)}} = \mathbb{E}_{ \mathcal{D}, \tilde{\mathcal{D}} } \frac{1}{K} \sum\_{k=1}^{K}
\frac{1}{|\tilde{\mathcal{D}}^{(k)} |} \left[ \left \| \left \| \tilde{\mathbf{y}}^{(k)} - \tilde{\mathbf{A}}^{(k)} \hat{\mathbf{x}}^{\text{(1st)}} \right \| \right\|_2^2  \right], $$
and equation (8) could be

$$\epsilon^{\text{(2nd)}} = \mathbb{E}_{ \mathcal{D}, \tilde{\mathcal{D}} }
\frac{1}{|\tilde{\mathcal{D}}^{(1)} |} \left[ \left \| \tilde{\mathbf{y}}^{(1)} - \tilde{\mathbf{A}}^{(1)} ( \kappa \hat{\mathbf{x}}^{\text{(1st)}}  + \hat{\mathbf{x}}^{\text{(2nd)}}  )\right\|_2^2  \right].$$

How would the change in the definition of the generalization error affect the results and bounds (if at all)? Alternately, it is good to discuss why this is an appropriate notion of generalization?
- (W3) Given the simplified hyperparameter search strategies proposed in this paper based on the analysis, it is not clear whether it has actually lead to smaller hyperparameter search spaces since there are baselines which reparameterize the hyperparameters to reduce one hyperparameter as well.
  - (W3.1) Given the Pretraining Lasso formulation with a total of 3 hyperparameters, $\lambda_1, \lambda_2, s$ with $\kappa \triangleq 1 - s$ and $\Delta\lambda \triangleq (1-s) \lambda_2 / s$, what computational advantage does Generalized Trans-Lasso provide if any since it also has to tune 3 hyperparameters?
  - (W3.2) It seems that a simple reformulation of the hyperparameters (similar to Pretraining Lasso) as $\kappa = 1 - s, \Delta \lambda = s \lambda_2 / (1 - s)$ span the spectrum of the $\kappa = 0$ and $\Delta \lambda = 0$ strategies, with $s = 0$ giving us the $\Delta\lambda = 0$ strategy, and the $s = 1$ giving us the $\kappa = 0$ strategy? This again reduces the number of hyperparameters then to 3 (instead of 4). And, with the strategies proposed by the theory in this paper, we still need to solve a 3 hyperparameter optimization problem (with either LO or GO). So what gain can Generalized Trans-Lasso provide over this?
  - (W3.3) For the real-data experiments, what are the error bars around the metrics in Table 1? Without that, it is not clear if the differences are significant.

Beyond the above, here are some less significant comments and questions:
- (C1) Is a large $N \to \infty$ limit where $N_k / N \to O(1)$ (so $N_k \to \infty$) an interesting usecase for practical sparse regression? Would the existing analysis be able to handle the case where $N_k / N \to 0$ while $N \to \infty$?
- (C2) The interpretation regarding noise levels and $\kappa = 0$ vs $\Delta \lambda = 0$ strategies, it is a bit unclear why the $\Delta \lambda = 0$ strategy (share actual feature vector) performs better that $\kappa = 0$ strategy (only share support information) in the high noise setting. As explained in 4th paragraph in Section 5 (page 9), "Consequently, it may be preferable to transfer more robust information, in this case the support configuration, rather than the noisy vector itself obtained in the first stage." Hence, in higher noise setting $\kappa = 0$ strategy would seem be the desirable one that just shares support information. What am I missing here?
- (C3) Figure 3 can be improved where we have the same heatmap scale between the different columns since we are supposed to be comparing the ratios $\epsilon / \epsilon_{LO}$ where $\epsilon \in \{ \min\lbrace \epsilon_{\Delta \lambda = 0}, \epsilon_{\kappa = 0} \rbrace, \epsilon_{\text{Pretrain}}, \epsilon_{\text{Trans}} \}$. Otherwise, it is hard to compare the performances between the methods.
- (C4) In page 2, step 1 (before equation (1)), should it be $\hat{\mathbf{x}}^{\text{(1st)}}$ in place of $\hat{\mathbf{x}}_1$?
- (C5) How are the two tasks different in the experiments in Section 4. Everything is Gaussian. Are the sampled non-overlapping $\mathcal{I}(k), k \in \{1, 2\}$ the only thing different? It would be good to clarify this in the experiment description.
- (C6) Where are the results for Trans-Lasso that is mentioned in the "Comparison with the original Trans-Lasso" paragraph on page 10? Is it from Figure 3?  It would be good to refer to the appropriate figure.

---

> ### Author Response · Authors · 2024-11-19
> **Response to Reviewer jR6M (1/3)**
>
> Thank you for investing your time in thoroughly reading our manuscript, and providing us with valuable comments.
>
> (W1.1) We have clarified the role of $t_1, t_2$ in the subsection 3.1 in the revised manuscript. The calculation in subsection 3.1 is to express the generalization error without using the normalized measure. Alternatively expressing it as a derivative of the logarithm allows us to conveniently evaluate this term. See also our reply to reviewer XUb3, and our reply to W1.2.
>
> (W1.2) Thank you for pointing this out; indeed, the discrepancy is confusing, and there should be $\exp(t_1 \cdots)$ or $\exp (t_2 \cdots)$ in front of $w_{\beta_1, \beta_2}$ in equation 13. To make our argument more generalized, and to address W1.4, we put instead $\exp (f(x_1, x_2; \mathcal{D}))$ for some appropriate function $f$.
>
> (W1.3) We have added a remark on this.
>
> (W1.4, 1.5) We admit the definition of the equations of state was misleading. The averaged quantities of random variables concerning the random optimization problems, which appear on the right hand side of equations of state, should be considered as a function of the parameters in question. $\Theta_1$ is then defined such that the left-hand side and the right hand side give equal quantities. Therefore, there is no bilevel structure in defining the equations of state; it is only a fixed-point equation, with the right-hand side given by a rather complicated function of $\Theta_1$ defined via expectation of random variables. We have clarified this by rewording definitions 1 and 2 accordingly.
>
> * (W1.4.1) Unfortunately, it is not easy to introduce $\Theta_1$ and $\Theta_2$ in a clear-cut way, since some parameters are purely auxiliary with little obvious interpretations. Instead, we have added an explanation on how these variables, as well as the equations of state, appear in the last paragraph of 3.1 in the revised manuscript. Additionally, we have added a new subsection 3.3 to address a subset of the parameters in $\Theta_1, \Theta_2$ which are straightforward to interpret; See also our reply to W1.6.
>
> * (W1.4.3) The existence of the solution to the equations of state is a highly non-trivial question, which can only be investigated numerically. Note that for the first stage, one can prove the existence of a unique solution using rigorous arguments (Thrampoulidis, Abbasi and Hassibi 2018; Miolane and Montanari 2021), although extending it to the second stage is out of the scope of our paper.
>
> * (W1.4.4) The comments are in general true, but in our case the variables $\mathsf{z}_1$ and $\mathsf{z}_2$ are Gaussian and the derivatives can be handled using Stein's lemma. Resultantly, the derivatives have explicit non-ambiguous formulae using the integrals over those variables. However, their explicit forms are notoriously complicated and we have chosen not to include them in the paper.
>
> (W1.6) Indeed, a subset of the parameters in $\Theta_1, \Theta_2$ have clear interpretations which are useful to explain why the generalization errors can be expressed as in Claim 1 and 2. We have added a new subsection 3.3 to explain all of this. We hope this clarifies the interpretation of the expressions.

---

> > ### Author Response · Authors · 2024-11-19
> > **Response to Reviewer jR6M (2/3)**
> >
> > (W2, 2-1) The difference between the definition of $\epsilon^{\rm (2nd)}$ given by us, and the one suggested by the reviewer is purely cosmetic, since it just yields the overall factor difference which has no impact on the hyperparameter selection.
> >
> > On the other hand, the difference on $\epsilon^{\rm (1st)}$ is influential. We, however, discuss that our definition is more appropriate in our setting. First of all, our definition of generalization error is compatible with the terms in our objective function in the training phase, implying that the same class distribution is assumed both in the training and the test phases, and that the test data is redrawn from the same generative model including the covariate matrix (the common and individual support model with random covariates in our case) with the same hyperparameters as the training data. Since we are not considering any *distribution shift* or *domain shift* in the present paper, there is no specific reason to change the class distribution in those two phases, meaning that each class $k$ can have a different impact on the error. Actually, our formula of $\epsilon^{\rm (1st)}$ derived from the replica method (equation (21)) shows that the class $k$'s contribution is proportional to $\alpha^{(k)}$. If the reviewer-suggesting definition is adopted, this replica formula would become equation (21) with all $\alpha^{(k)}$ fixed equal. We think this definition is more suitable if the test data has an equal weight on all the classes, but in such a situation it may be better also to re-weight the terms in our objective function in the training phase to balance the contributions among classes (and actually such techniques are adopted in imbalanced classification), leading to another scrutinization in the algorithm. Considering that our primary objective in this paper is to elucidate the hyperparameter dependence of the two algorithms (Pretraining Lasso and Trans-Lasso) and those algorithms have no such scrutinization, we think our definition is more appropriate for our purpose.
> >
> > Meanwhile, your suggestion and the consideration so far highlight the potential extensibility of these algorithms and our analysis, which we find highly intriguing and gives a potential future work. Thank you for your insightful comment.
> >
> >
> > (W3, 3-1) First of all, the algorithm extended to incorporate four hyperparameters is what we define as Generalized Trans-Lasso. This proposition of the algorithm is the first result of this paper. By analyzing this Generalized Trans-Lasso algorithm using the replica method, we could show that the two well-known algorithms, Pretraining Lasso and Trans-Lasso, are suboptimal. This is the second result. By further examining the replica result, we found that one of the two hyperparameters in Generalized Trans-Lasso are mutually redundant with respect to the generalization error, implying that the better generalization can be obtained in a *comparable* computational cost in hyperparameter search with that of Pretraining Lasso. This is the third result. Therefore, we do not claim any computational advantage over Trans-Lasso or Pretraining Lasso.
> >
> > (W3.2) Compared to our recommended approach of trying both the $\kappa = 0$ and $\Delta \lambda = 0$ strategies, your proposed reformulation requires a search only over $s$, and thus its computational cost would be roughly half of ours. However, your suggestion inevitably leaves certain parameter regions unexplored compared to ours and hence it would become suboptimal.
> >
> > (W3.3) We apologize for not providing the statistical errors. We have added them in table 1.

---

> > > ### Author Response · Authors · 2024-11-19
> > > **Response to Reviewer jR6M (3/3)**
> > >
> > > (C1) The limit $N_k /N \to O(1), M_k / N \to O(1)$ is referred to as the high-dimensional limit, which has been investigated thoroughly in the past decade not only in the context of theory, but also in the context of establishing guarantees on efficient algorithms. The limit of $N_k/ N \to 0$ as $N_k \to \infty$ can be handled similarily using the replica method, but we note that this would lead to a completely different analysis, see for instance Okajima et. al. 2023, or
> > >
> > > * Meng, Xiangming, Tomoyuki Obuchi, and Yoshiyuki Kabashima. "Ising Model Selection Using $l_1$-Regularized Linear Regression: A Statistical Mechanics Analysis." NeurIPS 34 (2021): 6290-6303.
> > >
> > > (C2) In this paragraph, we do not make an argument that the support information is robust in any case. Rather, we argue that the support configuration could be more informative compared to the first stage regressor in the low-SNR region in green, which will always induce some noise into the second optimization problem. This induced noise can be adversary to the extent that one should never transfer $x^{(\rm 1st)}$, which is the $\kappa = 0$ strategy we propose.
> > >
> > > (C3) Figure 3 is introduced to visualize how Pretraining Lasso, Trans-Lasso, and the generalized Trans-Lasso perform compared to the locally optimal strategy, and not to directly compare all three against each other. Note that as given in the caption of figure 3, figures with shared color bars are given in Appendix B, but it is much more difficult to see how close the generalized Trans-Lasso can achieve locally optimal performance.
> > >
> > > (C4) Thank you for pointing this out. We have corrected this in the revised manuscript.
> > >
> > > (C5) The experimental setup is thoroughly explained in the second paragraph in section 4. We consider 3 cases of target dataset size, $\alpha^{(1)} = 0.2, 0.4, 0.6$, with the auxiliary dataset given by $\alpha^{(2)} = 0.8$. Among all three dataset sizes, we further consider two cases for the common support size. Please let us know if there are any points ambiguous in the description.
> > >
> > > (C6) By this paragraph, we implied that the generalization error of the Trans-Lasso, which is basically the $\Delta \lambda = 0$ strategy with $\kappa = 1$ fixed, is bounded from below by that of the $\Delta \lambda = 0$ strategy. The original manuscript's paragraph title was "Comparison with the original Trans-Lasso" but we admit this is misleading. We have corrected it to "Implications on the original Trans-Lasso." in the revised manuscript.

---

### Review · Reviewer_nkcD · 2024-10-29

**Summary Of Contributions:**

The authors study transfer learning with the lasso in a high-dimensional linear regression setting via the replica method. Specifically, they consider a two-stage training setting (pretraining and then training on the target dataset) which generalizes two previous methods (Trans-Lasso and Pretraining Lasso) to allow for a more flexible transfer of two possible types of information from the first stage. The two types of information transfer are the support of the linear regression coefficients, or the coefficients themselves.

Using the replica method, they derive a set of equations whose solution gives the generalization error under different hyperparameter settings. They confirm a good agreement between the replica method predictions and empirical results on synthetic data which matches their theoretical setup, then use this to give guidelines on hyperparameter tuning and when one should prefer to transfer one type of information (support or coefficients) vs. the other. Finally, they explore the effect of the two different types of transfer on the IMDb dataset to see if their insights generalize to real data.

**Audience:**

Yes

**Claims And Evidence:**

No

**Requested Changes:**

1. Please provide more explanation of the connection between the first part of the paper/insights developed from the replica method and the practical results on the IMDb dataset. Is there some aspect of the theory and the properties of the IMDb dataset that allows us to predict the results of Fig. 4 a priori? If not, how would you suggest we apply the theoretical insights in practice?

2. Please clarify the question on equations (11) & (12) from the weaknesses section.

**Strengths And Weaknesses:**

# Strengths

The paper is generally well-written. The authors do a good job of explaining different aspects of the replica method, specifically its limitations (e.g., the fact that it is a heuristic) as well as how its output (which is a set of equations whose behavior is not obvious upon inspection) can be used to make useful predictions. The topic should also be of interest to some members of the TMLR community. The pre-training/fine-tuning paradigm has gained significant popularity and influence recently, and while the present paper studies an admittedly highly stylized version of this problem, developing theoretical insights in simplified settings can be an important step towards understanding the more complicated practical settings.

# Weaknesses

I'm not fully convinced that the relevance of the theoretical results for the practical setting studied in the paper has been substantiated. In their case study on the IMDb dataset, the authors show that tuning $\Delta \lambda$ has very little effect on the test error. There are some settings where a similar effect is observed in theory (e.g., the bottom-left plot in Fig. 1), but it's not clear why we should expect the IMDb dataset to fall into this category. At present, it seems that the only connection between the theoretical results and practice is that for the IMDb dataset, the $\Delta \lambda = 0$ strategy happens to work well. In order to claim that the theory gives practically relevant insight, it seems to me that the authors should show how one can use the qualitative insights from the theory and some properties of the IMDb dataset to predict this behavior a priori, without first doing an extensive hyperparameter search.

In equations (11) & (12), the integral is taken with respect to the unnormalized measure $w_{\beta_1, \beta_2}(x_1, x_2; \mathcal{D})$ rather than the probability measure $P_{\beta_1, \beta_2}$. Is this a typo, or is there some reason that we can use the unnormalized measure instead? Also, the intuition for why we take the limit of $\beta_1, \beta_2 \to \infty$ was clear; can similar intuition be used to explain the limits $t_1, t_2\to 0$?

---

> ### Author Response · Authors · 2024-11-19
> **Response to Reviewer nkcD**
>
> Thank you for your constructive comments.
> >Please provide more explanation of the connection between the first part of the paper/insights developed from the replica method and the practical results on the IMDb dataset. Is there some aspect of the theory and the properties of the IMDb dataset that allows us to predict the results of Fig. 4 a priori? If not, how would you suggest we apply the theoretical insights in practice?
>
> Admittedly, it is not easy to directly connect theory and real-data experiment, because one cannot be certain on the underlying model of the dataset. Our model is linear, but most of real-world datasets are expected to be non-linear, and also whether the assumptions of the common and individual support model are satisfied or not is nontrivial. However, there would exist datasets that satisfy such assumptions relatively well, and in such cases, our theoretical insights are expected to be useful. The experiment on the IMDb dataset was intended to illustrate the existence of such nontrivial datasets, rather than to make a claim that our theoretical findings universally hold in real-world scenarios. Fortunately, the experimental result on the IMDb dataset aligns with our theoretical finding that one of the two hyperparameters has little impact on generalization performance. This consistency shows that our theory is not merely a vacuous construct. In this way, we think it is more practical to judge whether the theory aligns with the real-world scenarios based on a posteriori outcomes through experiments.
>
> On the other hand, it is certainly desirable if we could a priori judge how the dataset and theory align. This point may be better addressed by providing well-controlled experiments where the profile of the data, such as the noise level, dataset size and support overlap, are under control. As such an experiment, we have added an experimental result on a compressive imaging task using linear random observations of MNIST images. The result demonstrates that when the signal-to-noise ratio of the measurement channel is small, the support information is crucial to improve on generalization. The results in this semi-artificial setup, which is a rather versatile one in the field of compressed sensing [ref1, ref2], demonstrate the qualitative agreement with our theoretical findings and provide some intuitive explanations when our two proxy strategies outperform the other.
>
> * [ref1] Baraniuk, Richard. "A lecture on compressive sensing." IEEE Signal processing magazine 24.4 (2007).
> * [ref2] Tan, Jin, Yanting Ma, and Dror Baron. "Compressive imaging via approximate message passing with image denoising." IEEE Transactions on Signal Processing 63.8 (2015): 2085-2092.
>
> >In equations (11) & (12), the integral is taken with respect to the unnormalized measure $w_{\beta_1, \beta_2}(x_1, x_2 ; \mathcal{D})$ rather than the probability measure $P_{\beta_1, \beta_2}$. Is this a typo, or is there some reason that we can use the unnormalized measure instead? Also, the intuition for why we take the limit of $\beta_1, \beta_2 \to \infty$ was clear; can similar intuition be used to explain the limits $t_1, t_2 \to 0$?
>
> They are not typo. Differentiation w.r.t. $t_1,t_2$ of $\log (\cdot)$ automatically yields the normalization. Meanwhile, the limits $t_1,t_2\to 0$ are taken by the same reason as that for the cumulant generating function (CGF): When computing a cumulant from CGF, we first differentiate CGF w.r.t its argument for the desired times for the cumulant to be computed, and take the zero limit of the argument, to avoid tilting the probability measure. Actually, the logarithm of the normalization is nothing but CGF in our problem setup. We have clarified these points in the revised manuscript.

---

### Review · Reviewer_XUb3 · 2024-10-30

**Summary Of Contributions:**

Authors study an algorithm, the Generalized Trans-Lasso, in a high-dimensional setting of Transfer learning via an asymptotic analysis using the replica method to address the issue that selection strategies for these hyperparameters, as well as the impact of these choices on other algorithm’s (such as Trans-Lasso and Pretraining Lasso) performance, have been largely unexplored.
They provide an asymptotic analysis as well as synthetic and real world data experiments.

**Audience:**

Yes

**Broader Impact Concerns:**

-

**Claims And Evidence:**

Yes

**Requested Changes:**

There are a lot of variables which makes it hard to follow especially if one is not familar with Transfer learning and it would be nice if there was a page with a definition and expanation of all variables (can be in the appendix).

Last line page 10: Results \*are\* consistent...

**Strengths And Weaknesses:**

strengths:

-both theoretic analysis as well as experiments are provided;

-authors work on a popular problem;

-the quality of the writeup is good.

weaknesses:

-strong assumptions on the data, among others the true feature vectors are also assumed to have i.i.d. standard Gaussian entries;

-in the real world dataset the hyperparameter $\Delta \lambda$ does not have a significant influence. Is there also a real world dataset where tuning $\Delta \lambda$ improves the error?

---

> ### Author Response · Authors · 2024-11-19
> **Response to Reviewer XUb3**
>
> Thank you for your generous feedback.
> >-in the real world dataset the hyperparameter
> $\Delta \lambda$
>  does not have a significant influence. Is there also a real world dataset where tuning $\Delta \lambda$
>  improves the error?
>
> In addition to the IMDb dataset experiment, we have added results of a compressive imaging task on the MNIST dataset to the updated manuscript. Reflecting on the insights obtained from synthetic experiments, we can see that when the measurements of the image exhibit low noise, the support information is also crucial in improving retrieval performance.
>
> >There are a lot of variables which makes it hard to follow especially if one is not familar with Transfer learning and it would be nice if there was a page with a definition and expanation of all variables (can be in the appendix).
>
> Thank you for your comment. We have added a table after Problem Setup for some notations which can be difficult to follow, to improve readability.

---

### Author Response · Authors · 2024-11-19
**General Response**

We thank all the reviewers for their time in reviewing our work. In response to everyone's feedback, we have updated our paper, where the main changes are in red. They are summarized in the following:

* Added paragraph on the choice of generalization error after equation (8).
* Clarified the role of $t_1, t_2$ in subsection 3.1.
* Reworded Definition 1 and 2 in a more clear way.
* Added interpretation on certain parameters in $\Theta_1, \Theta_2$ in new subsection 3.3.
* Added experimental results on a compressed imaging task using MNIST images.

---

### Decision · Action_Editor_uQcM · 2024-12-22

**Recommendation:** Accept as is

**Comment:**

This paper focuses on the problem of transfer learning in sparse high-dimensional linear regression, and studies the hyperparameter selection aspect of some recent two-stage regression schemes. Proposing a generalization of two existing schemes, the paper pursues a direct relationship between the generalization error of this scheme (for both the stages) and the selected hyperparameters. For this purpose, the authors perform an asymptotic analysis under various assumptions (Gaussian design matrices and feature vectors) utilizing tools from statistical physics (the replica method), establishing a systems of equations whose solution lead to closed-form analytic expressions for the generalization error utilizing problem dependent quantities. Then the authors highlight how the estimated generalization error in a finite data setting closely matches the asymptotic estimate (given that the data satisfy the required assumptions and that various data dependent quantities are known), thereby motivating the asymptotic analysis. The authors address reviewers' concerns well in their rebuttal. Thus, I would like to recommend accept as is.

**Audience:**

Yes

**Claims And Evidence:**

Yes